# Classifier Clustering and Feature Alignment for Federated Learning under Distributed Concept Drift

**Junbao Chen**
Beijing Institute of Technology
junbaochen@bit.edu.cn

**Jingfeng Xue**
Beijing Institute of Technology
xuejf@bit.edu.cn

**Yong Wang**
Beijing Institute of Technology
wangyong@bit.edu.cn

**Zhenyan Liu***
Beijing Institute of Technology
zhenyanliu@bit.edu.cn

**Lu Huang**
Beijing Institute of Technology
luhuang@bit.edu.cn

## Abstract

Data heterogeneity is one of the key challenges in federated learning, and many efforts have been devoted to tackling this problem. However, distributed concept drift with data heterogeneity, where clients may additionally experience different concept drifts, is a largely unexplored area. In this work, we focus on real drift, where the conditional distribution $P(\mathcal{Y}|\mathcal{X})$ changes. We first study how distributed concept drift affects the model training and find that local classifier plays a critical role in drift adaptation. Moreover, to address data heterogeneity, we study the feature alignment under distributed concept drift, and find two factors that are crucial for feature alignment: the conditional distribution $P(\mathcal{Y}|\mathcal{X})$ and the degree of data heterogeneity. Motivated by the above findings, we propose FedCCFA, a federated learning framework with classifier clustering and feature alignment. To enhance collaboration under distributed concept drift, FedCCFA clusters local classifiers at class-level and generates clustered feature anchors according to the clustering results. Assisted by these anchors, FedCCFA adaptively aligns clients' feature spaces based on the entropy of label distribution $P(\mathcal{Y})$, alleviating the inconsistency in feature space. Our results demonstrate that FedCCFA significantly outperforms existing methods under various concept drift settings. Code is available at https://github.com/Chen-Junbao/FedCCFA.

## 1 Introduction

Federated Learning (FL) [28] is an emerging privacy-preserving machine learning paradigm that allows multiple clients to collaboratively train a global model without sharing their raw data. In FL, clients train the models on their local data and send the updated models to the server for aggregation. Driven by the growing need for privacy protection, FL has been widely applied in various real-world scenarios [8, 16, 30, 44].

One significant challenge in FL is data heterogeneity, which denotes the discrepancies in the data distributions across clients. Such discrepancies can hinder the convergence of the global model.

---

*Corresponding author.

38th Conference on Neural Information Processing Systems (NeurIPS 2024).

However, existing works neglect a real-world setting where *data heterogeneity* and *distributed concept drift* simultaneously exist. Unlike conventional concept drift in centralized machine learning [11, 25, 26, 38], distributed concept drift [19] involves multiple clients experiencing different concept drifts at different times. For example, for the same medical image, diagnoses can vary among doctors (i.e., concept drift across clients), and even a doctor may offer different diagnoses at different times (i.e., concept drift across times). Moreover, the distribution of medical images varies across hospitals (i.e., data heterogeneity). This setting significantly degrades the performance of most FL methods, especially those using a single global model, because the model cannot provide different outputs for the same input. A similar research problem is multistream classification [4, 15, 46], where a sampling bias may exist between the distributions represented by source stream and target stream. Different from this problem, distributed concept drift focuses on the changing conditional distribution $P(\mathcal{Y}|\mathcal{X})$ across clients and over time. For each client at any round, training and test data distribution are assumed to be similar.

Several recent works have recognized the concept drift problem in FL. However, as discussed above, these single-model solutions [2, 3, 5, 14, 32] are ill-suited for distributed concept drift. FedDrift [19] considers distributed concept drift and employs multiple models to address this problem. However, FedDrift significantly increases the computation overhead for its distance measure and the storage overhead for multiple entire models. In addition, since the data heterogeneity can affect the loss estimation, FedDrift may fail to merge the models under the same concept. Experimental details are provided in Appendix C.7.

To tackle distributed concept drift, we analyze how it affects the model training in FL. In this work, we focus on *real drift* [11, 38] in concept drift, where the conditional distribution $P(\mathcal{Y}|\mathcal{X})$ changes. When $P(\mathcal{Y}|\mathcal{X})$ varies across clients, the representation should not be affected, as the marginal distribution $P(\mathcal{X})$ is invariant. However, as shown in Figure 1, when vanilla FedAvg is adopted and distributed concept drift occurs at round 100, the Frobenius norm of representation update increases drastically, suggesting that the drift leads to large gradients in the representation. Furthermore, the accuracy of vanilla FedAvg drops rapidly and cannot recover to the accuracy before the drift, indicating the unsuitability of the single-model solution.

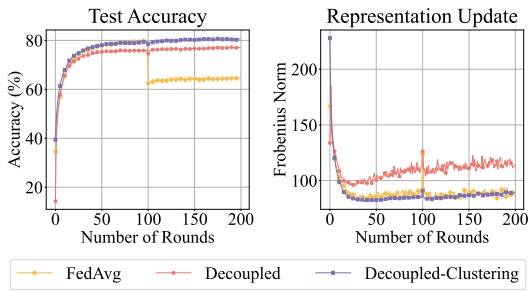

Figure 1: The impact of distributed concept drift on model training. Distributed concept drift occurs at round 100. Decoupled: classifier-then-extractor learning method. Decoupled-Clustering: Decoupled method with classifier clustering.

Motivated by the observations and analyses above, we decouple the network into an extractor and a classifier, and first train the classifier while fixing the extractor. After the classifier learns new conditional distribution, small gradients will be back-propagated to the extractor, and then we train the extractor. This decoupled method can effectively adapt to distributed concept drift, demonstrated by the much higher accuracy than FedAvg in Figure 1. The small gradient norm of representation at drift round also indicates the small gradients back-propagated from classifier. However, some clients may share similar $P(\mathcal{Y}|\mathcal{X})$, and pure decoupled methods neglect the fine-grained collaboration between local classifiers. This will make each client's local classifier overfit to its local data. To enhance generalization performance, we develop a class-level classifier clustering method. Our clustering method separates the classifier for each class (referred to as class classifier), and then aggregates clients' class classifiers trained under the same conditional distribution $P(\mathcal{Y}|\mathcal{X})$. The clients under the same conditional distribution share the aggregated class classifiers. This aggregation reduces the bias introduced by any single client's data, contributing to improved generalization performance. As shown in Figure 1, with the benefit of classifier clustering, the generalization performance is further improved, demonstrated by higher accuracy and smaller gradient norm. To remedy the data heterogeneity under distributed concept drift, we propose an adaptive feature alignment method, which aligns the feature spaces of the clients with the same conditional distribution $P(\mathcal{Y}|\mathcal{X})$ and adjusts the alignment weight according to the entropy of label distribution.

To summarize our contributions:

1. We explore the impact of distributed concept drift on FL training and propose a class-level classifier clustering approach that not only adapts to this drift but also enhances generalization performance (Section 4.1).

2. We propose clustered feature anchors to achieve feature alignment under distributed concept drift and propose an adaptive alignment weight to prevent severe data heterogeneity from impeding main task learning (Section 4.2).

3. We propose FedCCFA, a federated learning framework with classifier clustering and feature alignment (see Section 4.3). In Section 5, we conduct extensive experiments and empirical results demonstrate that FedCCFA can effectively adapt to distributed concept drift under data heterogeneity and significantly outperforms existing methods.

## 2   Related work

**Concept drift in FL.**   Concept drift has been extensively studied in centralized machine learning [11, 12, 25, 26, 38]. Several recent works have recognized the concept drift problem in FL and proposed methods to tackle this issue, including regularization [5], continual learning [3, 14] and adaptive learning rate [2, 32]. These works assume that the conditional distributions $P(\mathcal{Y}|\mathcal{X})$ of all clients' dataset are the same, so only a single global model is trained. However, as proposed in [19] and discussed in Section 1, distributed concept drift may exist in FL setting and the single-model solution fails to adapt to this drift. To address this problem, FedDrift [19] creates new models based on drift detection and adaptively merges models by hierarchical clustering. However, FedDrift still faces some challenges: 1) incorrect model merging caused by data heterogeneity; 2) high computation and communication overhead for measuring cluster distance; 3) high storage cost for maintaining multiple global models.

**Data heterogeneity in FL.**   Data heterogeneity hinders fast convergence when using vanilla FedAvg [28]. In this work, we focus on the heterogeneity of label distributions (referred to as label distribution skew). To alleviate this issue, previous works either focus on local training [1, 20, 22, 24, 29] or global aggregation [17, 18, 33, 39–41]. To further study how data heterogeneity affects FL, several recent works have focused on the representation of model, such as dimensional collapse [36] and inconsistent feature spaces [43, 45, 48, 50]. To align clients' feature spaces, FedFA [50] and FedPAC [43] regularize the $\ell_2$ distance between local features and global anchors. To promote more precise alignment, FedFM [45] uses a contrastive-guiding method to further maximize the distance between the feature and non-corresponding anchors. However, severe data heterogeneity may significantly increase the loss of feature alignment, hindering the model convergence.

**Clustered FL and personalized FL.**   Some clustered FL methods group clients with the same data distribution into a cluster, which can also adapt to distributed concept drift. To group clients, several works measure the similarity based on gradient information [9, 10, 35] or training loss [13, 27]. However, these methods face the following challenges: 1) unknown number of clusters; 2) large computation and communication overhead for estimating the training loss; 3) considerable storage cost for multiple global models. Personalized FL methods [6, 23, 31, 37, 43, 49] are also robust to distributed concept drift, since each client trains a local model for its local data distribution. However, most personalized FL methods neglect the classifier collaboration among the clients with similar data distributions, limiting the performance of models. Different from personalized FL approaches, this work focuses on generalized FL, aiming to enhance generalization performance.

## 3   Problem formulation

We consider a FL setting where there are $K$ clients and a central server. The $k$-th client has a local dataset $D_k$ with data distribution $P_k(\mathcal{X}, \mathcal{Y})$, where $\mathcal{X}$ is the input space and $\mathcal{Y}$ is the label space with $C$ classes. Let $\ell(\mathbf{w}; \mathbf{x}, y)$ be the task-specific loss function associated with model $\mathbf{w}$ and data sample $(\mathbf{x}, y)$. The global objective of FL can be formulated as:

$$\min_{\mathbf{w} \in \mathbb{R}^d} \{F(\mathbf{w}) := \sum_{k=1}^{K} p_k F_k(\mathbf{w})\} \tag{1}$$

where $p_k$ is the aggregation weight of the $k$-th client and $F_k(\mathbf{w}) := \mathbb{E}_{(\mathbf{x},y)\sim D_k}[\ell_k(\mathbf{w};\mathbf{x},y)]$ is the local objective.

To distinguish different types of concept drift, we decompose the joint distribution $P(\mathcal{X},\mathcal{Y})$ as: $P(\mathcal{X},\mathcal{Y}) = P(\mathcal{X})P(\mathcal{Y}|\mathcal{X}) = P(\mathcal{Y})P(\mathcal{X}|\mathcal{Y})$. In this work, we focus on the *real drift* in concept drift. Real drift means that the conditional distribution at round $t$ may be different from that at the previous round, i.e., $P^{(t)}(\mathcal{Y}|\mathcal{X}) \neq P^{(t-1)}(\mathcal{Y}|\mathcal{X})$. In addition, under distributed concept drift, this conditional distribution may vary across clients, i.e., $P_i^{(t)}(\mathcal{Y}|\mathcal{X}) \neq P_j^{(t)}(\mathcal{Y}|\mathcal{X})$.

To address distributed concept drift, we decouple the model parameterized by $\mathbf{w} = \{\boldsymbol{\theta}, \boldsymbol{\phi}\}$ into a feature extractor $f_{\boldsymbol{\theta}}$ parameterized by $\boldsymbol{\theta}$ and a classifier $f_{\boldsymbol{\phi}}$ parameterized by $\boldsymbol{\phi}$. Given a sample $(\mathbf{x}, y) \in \mathcal{X} \times \mathcal{Y}$, the feature extractor $f_{\boldsymbol{\theta}} : \mathcal{X} \rightarrow \mathcal{Z}$ maps the input $\mathbf{x}$ into a feature vector $\mathbf{z} = f_{\boldsymbol{\theta}}(\mathbf{x})$ in the feature space $\mathcal{Z}$, and then the classifier $f_{\boldsymbol{\phi}} : \mathcal{Z} \rightarrow \mathcal{Y}$ maps the feature vector $\mathbf{z}$ to the label space $\mathcal{Y}$. All clients share the feature extractor and each client maintains its local classifier. To align clients' feature spaces, we introduce a regularization term $G_k(\boldsymbol{\theta}; \mathcal{A}_k)$ into the local objective function. Here, $\mathcal{A}_k$ represents a form of global information that client $k$ uses to align its feature space. Let $\boldsymbol{\Phi} = \{\boldsymbol{\phi}_1, \boldsymbol{\phi}_2, \ldots, \boldsymbol{\phi}_k\}$ denote the set of all clients' local classifier parameters and $\boldsymbol{\theta}$ denote the parameters of shared feature extractor. The global objective can be reformulated as:

$$\min_{\boldsymbol{\theta},\boldsymbol{\Phi}}\{F(\boldsymbol{\theta},\boldsymbol{\Phi}) := \sum_{k=1}^{K} p_k[F_k(\boldsymbol{\theta},\boldsymbol{\phi}_k) + \lambda G_k(\boldsymbol{\theta};\mathcal{A}_k)]\} \tag{2}$$

where $\lambda$ is the weight of feature alignment.

## 4 Methodology

FedCCFA decouples the network into a feature extractor $f_{\boldsymbol{\theta}}$ and a classifier $f_{\boldsymbol{\phi}}$. In FedCCFA, the classifier is the last layer of the model (i.e., a linear classifier), and the feature extractor is composed of the remaining layers. In this section, we first present two methods used in FedCCFA and then describe the design of FedCCFA.

### 4.1 Classifier clustering

For better classifier collaboration, we introduce a new method for client clustering. Specifically, all clients share the same extractor and train their personal classifiers. Given the identical dimension reduction $\mathbf{z} = f_{\boldsymbol{\theta}}(\mathbf{x})$, the personal classifier learns the local data distribution. Therefore, the linear classifiers with similar weights exhibit similar data distributions, especially the conditional distribution $P(\mathcal{Y}|\mathcal{X})$. Different from existing clustered FL methods, *FedCCFA directly uses the classifier weights for clustering*, which significantly reduces the computation and communication cost.

However, if using the weights of local classifiers, clustering may be disturbed by two factors: (1) variation in the classifier weights before training, and (2) class imbalance. For the first one, since clients' classifiers are different before local training, the classifiers trained under the same conditional distribution may differ significantly, which can lead to wrong clustering results. For the second one, the classifier can be easily biased towards head classes with massive training data, so the classifiers trained under the similar marginal distributions $P(\mathcal{X})$ may be grouped, although their conditional distributions $P(\mathcal{Y}|\mathcal{X})$ are different.

**Balanced classifier training.** To address the above two problems, before local classifier training, FedCCFA trains a balanced classifier and uses it for classifier clustering. Specifically, to ensure that the classifier weights before training are identical, each selected client $k \in \mathcal{I}^{(t)}$ updates its classifier $\boldsymbol{\phi}_k^{(t)}$ by the initial global classifier $\boldsymbol{\phi}^{(0)}$. Then, to address class imbalance, client $k$ samples a balanced batch $b_k^{(t)}$ from $D_k^{(t)}$ to train its classifier $\boldsymbol{\phi}_k^{(t)}$ while fixing its extractor $\boldsymbol{\theta}_k^{(t)}$:

$$\boldsymbol{\phi}_k^{(t)} \leftarrow \boldsymbol{\phi}_k^{(t)} - \eta_{\boldsymbol{\phi}}\nabla_{\boldsymbol{\phi}}F_k(\boldsymbol{\theta}_k^{(t)}, \boldsymbol{\phi}_k^{(t)}) \tag{3}$$

where $\eta_{\boldsymbol{\phi}}$ denotes the learning rate for classifier.

After $s$ training iterations, client $k$ saves the balanced classifier $\hat{\boldsymbol{\phi}}_k^{(t)}$ and sends it to the server after local training. Note that, to reduce additional computation cost, we randomly select 5 samples for

each class $c \in [C]$ (i.e., the balanced batch size $|b_k^{(t)}|$ is $5 * C$), and set training iterations $s$ to a small value.

**Class-level clustering.** Unlike FedPAC [43] that uses *entire* classifier for collaboration, FedCCFA conducts classifier clustering at *class-level*. Specifically, after receiving all balanced classifiers $\{\hat{\phi}_k^{(t)}\}$, the server separates these classifiers for each class (referred to as class classifier). Then, for each class $c$, the server measures the class-level distance $\mathcal{D}_c(i, j)$ between client $i$ and client $j$ by their class classifiers $\hat{\phi}_{i,c}^{(t)}$ and $\hat{\phi}_{j,c}^{(t)}$, where $i, j \in \mathcal{I}^{(t)}$. To effectively measure distance between high-dimensional vectors, we exploit MADD [34] and realize a measure based on cosine distance:

$$\mathcal{D}_c(i,j) = \frac{1}{|\mathcal{I}^{(t)}| - 2} \sum_{q \in \mathcal{I}^{(t)} \setminus \{i,j\}} \left| Cos(\hat{\phi}_{i,c}^{(t)}, \hat{\phi}_{q,c}^{(t)}) - Cos(\hat{\phi}_{j,c}^{(t)}, \hat{\phi}_{q,c}^{(t)}) \right|, \quad \forall i, j \in \mathcal{I}^{(t)} \quad (4)$$

where $Cos(\mathbf{a}, \mathbf{b})$ denotes the cosine distance between vector $\mathbf{a}$ and $\mathbf{b}$.

After getting the distance matrix $\mathcal{D}_c$, DBSCAN clustering algorithm is used to get the class-level clusters $\mathcal{S}_c^{(t)}$. For each cluster $\mathcal{S}_{m,c}^{(t)} \in \mathcal{S}_c^{(t)}$, clustered class classifier is aggregated as:

$$\bar{\phi}_{m,c}^{(t)} = \frac{1}{|\mathcal{S}_{m,c}^{(t)}|} \sum_{k \in \mathcal{S}_{m,c}^{(t)}} \phi_{k,c}^{(t)}, \quad \forall \mathcal{S}_{m,c}^{(t)} \in \mathcal{S}_c^{(t)} \quad (5)$$

where $|\mathcal{S}_{m,c}^{(t)}|$ denotes the number of clients in cluster $\mathcal{S}_{m,c}^{(t)}$. Each client $k \in \mathcal{S}_{m,c}^{(t)}$ updates its local class classifier $\phi_{k,c}^{(t)}$ by $\bar{\phi}_{m,c}^{(t)}$. Here we use uniform aggregation considering privacy concerns. More details about aggregation method are shown in Appendix C.6.

## 4.2 Adaptive feature alignment

To alleviate the inconsistency in feature spaces [45], we introduce a regularization term into the local objective function to align clients' feature spaces. Compared to existing alignment methods [43, 45, 50], our method is robust to two challenges: concept drift and the degree of data heterogeneity.

**Clustered feature anchors.** Existing feature alignment methods suppose that all clients' conditional distributions $P(\mathcal{Y}|\mathcal{X})$ are identical. Based on this, global feature anchors [45, 50] (also known as feature centroids [43]) are leveraged to align clients' feature spaces. However, under distributed concept drift scenario, the conditional distribution $P(\mathcal{Y}|\mathcal{X})$ may vary across clients, which can result in the differences in feature anchors. Therefore, clients require the global anchors that match their conditional distributions; otherwise, incorrect global anchors can mislead feature alignment. Motivated by this, we propose a feature alignment method using clustered feature anchors. Specifically, at round $t$, each client uses its feature extractor $\theta_k^{(t)}$ to compute the local feature anchor $a_{k,c}^{(t)}$ for each class $c$:

$$a_{k,c}^{(t)} = \frac{1}{|D_{k,c}^{(t)}|} \sum_{(\mathbf{x},c) \in D_{k,c}^{(t)}} f_{\boldsymbol{\theta}_k^{(t)}}(\mathbf{x}), \quad \forall c \in [C] \quad (6)$$

Then, with the help of our class-level clustering, we compute the global feature anchor $\mathcal{A}_{m,c}^{(t)}$ of class $c$ for each cluster $\mathcal{S}_{m,c}^{(t)}$:

$$\mathcal{A}_{m,c}^{(t)} = \frac{1}{|\mathcal{S}_{m,c}^{(t)}|} \sum_{k \in \mathcal{S}_{m,c}^{(t)}} a_{k,c}^{(t)}, \quad \forall \mathcal{S}_{m,c}^{(t)} \in \mathcal{S}_c^{(t)} \quad (7)$$

Finally, each client $k \in \mathcal{S}_{m,c}^{(t)}$ uses its global anchors $\mathcal{A}_k^{(t)} = \{\mathcal{A}_{m,c}^{(t)}\}_{c=1}^C$ to align its feature space during local training. FedCCFA leverages the contrastive-guiding loss proposed in FedFM [45], but uses clustered feature anchors. Let $sim(\mathbf{u}, \mathbf{v})$ denote the cosine similarity between $\mathbf{u}$ and $\mathbf{v}$. Then the alignment loss function for a sample $(\mathbf{x}, c)$ with label $c$ is defined as:

$$G_k(\boldsymbol{\theta}_k^{(t)}; \mathcal{A}_k^{(t)}) = -\log \frac{\exp(sim(f_{\boldsymbol{\theta}_k^{(t)}}(\mathbf{x}), \mathcal{A}_{k,c}^{(t)})/\tau)}{\sum_{i=1}^C \exp(sim(f_{\boldsymbol{\theta}_k^{(t)}}(\mathbf{x}), \mathcal{A}_{k,i}^{(t)})/\tau)} \quad (8)$$

where $f_{\boldsymbol{\theta}_k^{(t)}}(\mathbf{x})$ is the representation vector of input $\mathbf{x}$ and $\tau$ denotes a temperature parameter.

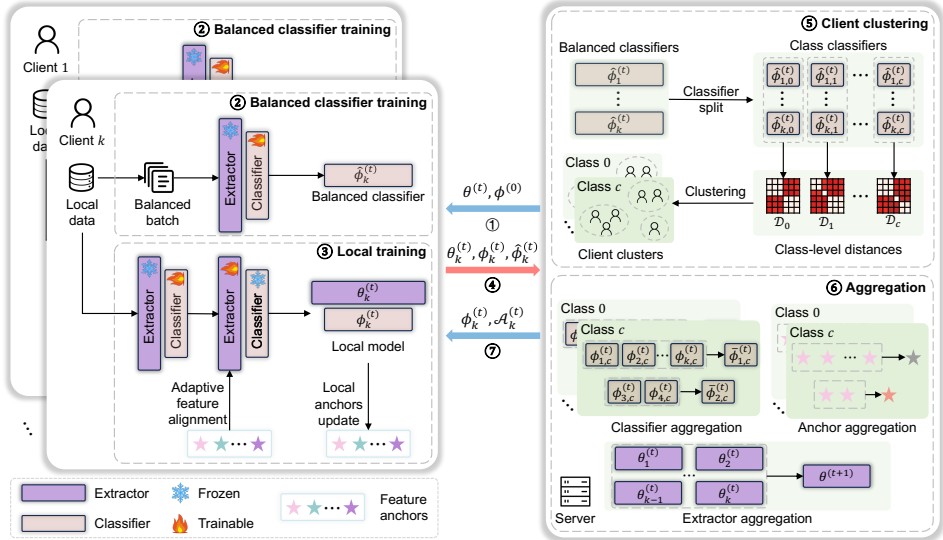

Figure 2: An overview of the proposed FedCCFA. Clients train balanced classifiers and local models, and then generate local anchors. The server performs client clustering with the help of balanced classifiers, and then aggregates local models and local anchors.

**Adaptive alignment weight.** In existing feature alignment methods [43, 45, 50], the alignment weight is fixed and all clients use the same weight. However, severe data heterogeneity will significantly increase the gradients of alignment term, impeding the main task learning. Experimental results are presented in Table 5. To balance the main task learning and feature alignment, it is essential to reduce the alignment weight under severe data heterogeneity. Note that, in this work, we focus on the label distribution skew and the degree of this heterogeneity can be reflected by the marginal distribution $P_k^{(t)}(\mathcal{Y})$. Based on this intuition, we leverage the entropy of marginal distribution $P_k^{(t)}(\mathcal{Y})$, denoted as $\mathcal{H}(P_k^{(t)}(\mathcal{Y}))$, to adaptively determine the alignment weight:

$$\boldsymbol{\theta}_k^{(t)} \leftarrow \boldsymbol{\theta}_k^{(t)} - \eta_{\boldsymbol{\theta}} \nabla_{\boldsymbol{\theta}} [F_k(\boldsymbol{\theta}_k^{(t)}, \boldsymbol{\phi}_k^{(t)}) + \frac{\mathcal{H}(P_k^{(t)}(\mathcal{Y}))}{\gamma} G_k(\boldsymbol{\theta}_k^{(t)}; \mathcal{A}_k^{(t)})] \tag{9}$$

where $\eta_{\boldsymbol{\theta}}$ denotes the learning rate for extractor and $\gamma$ is the scaling factor. For the round $T_s$ to start feature alignment, we use the empirical value $T_s = 20$ proposed in FedFM [45].

## 4.3 FedCCFA

We now present FedCCFA, a federated learning framework to adapt to distributed concept drift under data heterogeneity. The pipeline of FedCCFA is shown in Figure 2. The procedure of FedCCFA is formally presented in Algorithm 1 in Appendix A.

**Local training.** At each round $t$, each selected client $k \in \mathcal{I}^{(t)}$ updates its extractor $\boldsymbol{\theta}_k^{(t)}$ by the global extractor $\boldsymbol{\theta}^{(t)}$, and then starts local training procedure. Specifically, each client trains a balanced classifier (Equation 3) for client clustering. Then, it fixes its feature extractor and trains local classifier to adapt to concept drift. Finally, it trains its feature extractor (Equation 9). Since the classifier $\boldsymbol{\phi}_k^{(t)}$ learns the conditional distribution $P_k^{(t)}(\mathcal{Y}|\mathcal{X})$, concept drift will not lead to larges gradients in the representation. After local training, each client generates its local anchors for each class (Equation 6).

**Global aggregation.** After receiving the local parameters and local anchors, the server starts the aggregation procedure. Specifically, the server performs client clustering with the help of balanced classifiers. Then, the server aggregates all feature extractors:

$$\boldsymbol{\theta}^{(t+1)} = \frac{1}{\sum_{k \in \mathcal{I}^t} |D_k^{(t)}|} \sum_{k \in \mathcal{I}^t} |D_k^{(t)}| \boldsymbol{\theta}_k^{(t)} \qquad (10)$$

Finally, according to the clustering results, the server aggregates local classifiers and local anchors for each cluster.

## 5  Experiments

### 5.1  Experimental setup

**Datasets and models.**  We conduct experiments on three datasets, namely Fashion-MNIST [42], CIFAR-10 [21] and CINIC-10 [7]. We construct two different CNN models for Fashion-MNIST and CIFAR-10/CINIC-10, respectively. Details of datasets and models are provided in Appendix B.1.

**Baselines.**  We compare FedCCFA against the methods falling under the following categories: (1) single-model methods without drift adaptation: FedAvg [28], FedProx [24], SCAFFOLD [20] and FedFM [45]; (2) single-model methods with drift adaptation: Adaptive-FedAvg (shortened to AdapFedAvg) [2] and Flash [32]; (3) personalized FL methods: pFedMe [37], Ditto [23], FedRep [6], FedBABU [31] and FedPAC [43]; (4) clustered FL methods: IFCA [13] and FedDrift [19].

**Federated learning settings.**  We consider two FL scenarios: 20 clients with full participation and 100 clients with 20% participation. We use a widely considered non-IID setting [22, 39, 47]: for each class $c \in [C]$, we sample a probability vector $\mathbf{p}_c = (p_{c,1}, p_{c,2}, \ldots, p_{c,K}) \sim Dir_K(\alpha)$ and allocate a $p_{c,k}$ proportion of instances of class $c$ to client $k \in [K]$, where $Dir_K(\alpha)$ denotes the Dirichlet distribution with concentration parameter $\alpha$; smaller $\alpha$ means more unbalanced partition. To evaluate the adaptability to concept drift, each client's training set contains at least 5 samples per class.

**Concept drift settings.**  In this work, we focus on the distributed concept drift proposed in [19], where the conditional distribution $P(\mathcal{Y}|\mathcal{X})$ varies over time and across clients. We use three label swapping settings to simulate the conditional distribution changes across clients: for client $k \in [K]$, (1) class 1 and class 2 are swapped if $k\%10 < 3$; (2) class 3 and class 4 are swapped if $3 <= k\%10 <= 5$; (3) class 5 and class 6 are swapped if $k\%10 > 5$. We simulate three patterns of conditional distribution changes over time: (1) **Sudden Drift.** All label swapping settings occur at round 100; (2) **Incremental Drift.** Label swapping settings (1), (2) and (3) occur at round 100, 110 and 120, respectively; (3) **Reoccurring Drift.** All label swapping settings occur at round 100, and these settings occur again at round 150.

**Implementation details.**  We run 200 communication rounds for all experiments. To comprehensively evaluate the generalized performance and drift adaptability, for each client, we report the generalized accuracy of its model on *the whole test set*. In particular, the conditional distributions $P(\mathcal{Y}|\mathcal{X})$ of each client's training set and test set are the same at every round. For clustered and single-model methods, each client evaluates its global model; for the other methods, each client evaluates its personal model (or the combination of global extractor and personal classifier). Finally, we report the average accuracy across clients at the last round. For local training, we use SGD optimizer. The weight decay is 0.00001 and the SGD momentum is 0.9. For all datasets, we set local epochs $E = 5$. For the decoupled methods, extractor learning rate is 0.01 and classifier learning rate is 0.1; for the other methods, local learning rate is 0.01. For FedCCFA, scaling factor $\gamma$ is 20.0, iterations of balanced training $s$ is 5 and balanced batch size is $5 * C$ (5 samples per class). For classifier clustering, maximum distance $\epsilon$ and minimum samples used in DBSCAN algorithm are 0.1 and 1, respectively. More details of hyperparameters are provided in Appendix B.2.

### 5.2  Experimental results

We focus on three points in our experiments: (1) the generalization performance of FedCCFA; (2) the adaptability to distributed concept drift compared to existing methods; (3) the effects of our classifier clustering and adaptive alignment weight. Full experimental results are provided in Appendix C.

Table 1: Generalized accuracy under $Dir(0.5)$. The sample ratio is 100% and 20% for 20 clients and 100 clients, respectively. All results are averaged over 3 runs (mean ± std).

| Method | Fashion-MNIST | | CIFAR-10 | | CINIC-10 | |
| --- | --- | --- | --- | --- | --- | --- |
| | 20 clients | 100 clients | 20 clients | 100 clients | 20 clients | 100 clients |
| FedAvg | 90.57±0.18 | 90.22±0.33 | 78.56±0.20 | 74.05±0.72 | 62.23±0.28 | 58.28±0.21 |
| FedProx | 90.29±0.17 | 90.22±0.02 | 78.49±0.20 | 73.50±0.61 | 62.17±0.35 | 58.40±0.86 |
| SCAFFOLD | 90.33±0.04 | 87.72±0.19 | 75.84±0.71 | 59.01±0.74 | 58.92±0.12 | 47.69±0.35 |
| FedFM | 90.03±0.26 | 90.20±0.35 | 78.97±0.43 | 74.89±0.42 | 62.96±0.27 | 58.66±0.13 |
| AdapFedAvg | 90.42±0.25 | 90.39±0.11 | 78.28±0.38 | 73.91±0.69 | 61.96±0.30 | 58.80±0.42 |
| Flash | 90.19±0.05 | 89.94±0.30 | 78.01±0.61 | 75.13±0.23 | 61.11±0.43 | 60.30±0.25 |
| pFedMe | 85.06±0.32 | 84.81±0.31 | 63.08±1.28 | 51.49±0.08 | 43.75±0.07 | 41.80±0.49 |
| Ditto | 90.33±0.15 | 89.73±0.19 | 77.03±0.39 | 72.26±0.44 | 59.30±0.25 | 56.33±0.10 |
| FedRep | 81.70±0.10 | 80.39±0.30 | 64.08±0.30 | 52.19±1.50 | 43.63±0.38 | 38.48±0.12 |
| FedBABU | 76.83±1.66 | 80.69±0.54 | 58.99±0.25 | 55.42±0.21 | 40.74±0.21 | 41.43±0.26 |
| FedPAC | 85.01±0.39 | 87.61±0.26 | 67.96±0.40 | 67.77±0.36 | 43.83±0.69 | 47.50±0.38 |
| IFCA | 86.99±1.17 | 88.61±0.51 | 64.10±0.75 | 72.53±0.30 | 48.43±1.48 | 57.84±0.18 |
| FedDrift | 90.38±0.36 | 90.33±0.16 | 78.59±0.35 | 73.70±0.76 | 62.12±0.23 | 58.48±0.72 |
| FedCCFA | 89.81±0.36 | 89.74±0.18 | 78.38±0.56 | 74.44±0.26 | 61.13±0.25 | 58.70±0.33 |

Table 2: Generalized accuracy under sudden drift setting and $Dir(0.5)$. The sample ratio is 100% and 20% for 20 clients and 100 clients, respectively. All results are averaged over 3 runs (mean ± std).

| Method | Fashion-MNIST | | CIFAR-10 | | CINIC-10 | |
| --- | --- | --- | --- | --- | --- | --- |
| | 20 clients | 100 clients | 20 clients | 100 clients | 20 clients | 100 clients |
| FedAvg | 67.81±1.12 | 68.15±1.98 | 60.96±0.37 | 58.15±0.23 | 49.69±0.16 | 46.18±0.39 |
| FedProx | 69.25±0.60 | 68.47±0.19 | 61.25±0.20 | 57.68±0.11 | 49.45±0.40 | 46.11±0.59 |
| SCAFFOLD | 69.92±1.24 | 69.66±0.24 | 58.67±0.39 | 45.87±1.29 | 46.34±0.70 | 37.94±1.43 |
| FedFM | 68.48±0.79 | 69.12±0.34 | 60.61±0.27 | 57.51±0.35 | 50.25±0.11 | 46.23±0.81 |
| AdapFedAvg | 69.30±1.06 | 68.86±0.53 | 60.92±0.20 | 58.23±0.30 | 49.86±0.36 | 46.34±0.19 |
| Flash | 10.00±0.00 | 71.16±0.28 | 59.84±0.75 | 60.49±0.27 | 49.44±0.41 | 49.28±0.40 |
| pFedMe | 82.37±0.09 | 77.66±0.17 | 58.92±1.64 | 44.15±1.08 | 41.34±0.22 | 37.76±0.78 |
| Ditto | 78.51±0.82 | 79.62±0.31 | 67.45±0.01 | 63.35±0.67 | 51.06±0.26 | 48.40±0.34 |
| FedRep | 81.99±0.37 | 81.07±0.29 | 64.51±0.86 | 53.30±1.51 | 44.00±0.26 | 38.15±0.14 |
| FedBABU | 79.83±0.18 | 81.68±0.10 | 58.87±0.50 | 55.29±0.61 | 41.24±0.90 | 40.49±0.30 |
| FedPAC | 84.00±0.47 | 87.24±0.16 | 66.47±0.30 | 64.73±1.22 | 44.38±0.18 | 46.22±0.47 |
| IFCA | 87.02±0.06 | 88.25±0.21 | 61.24±0.32 | 57.31±0.44 | 44.04±2.00 | 45.43±0.22 |
| FedDrift | 78.95±1.30 | 87.75±0.04 | 48.17±2.43 | 57.26±0.36 | 34.23±0.04 | 45.60±1.07 |
| FedCCFA | **89.50±0.10** | **89.39±0.14** | **76.95±0.63** | **73.21±0.82** | **51.12±1.70** | **52.62±0.51** |

**Performance without concept drift.** First, we evaluate all methods under the no drift setting, and these results are the base performance compared against various concept drift settings. For each method, we run three trials and report the mean and standard deviation. Table 1 presents the generalized accuracy under $Dir(0.5)$. We observe that FedCCFA outperforms all decoupled methods (FedRep, FedBABU and FedPAC) and personalized FL methods (pFedMe and Ditto), indicating that FedCCFA enhances the generalization performance of local classifiers. Besides, the performance of IFCA under 20 clients with 100% participation is worse than the performance under 100 clients with 20% participation. This is because multiple global models are trained under similar data distribution (i.e., each global model is trained with much less data). In contrast, with the help of our classifier clustering, clients with similar data distribution will share the same classifier. Compared with single-model methods, FedCCFA achieves lower accuracy, which is attributed to decoupled training. Specifically, for decoupled methods (e.g., FedPAC, FedRep and our FedCCFA), training the classifier first might cause it to fit the initial features, resulting in gradient attenuation during backpropagation. This will restrict the subsequent extractor's training process and prevent it from optimally learning

Table 3: Ablation study on the clustering input. We run experiments under sudden drift setting and $Dir(0.5)$. We apply partial participation.

| Clustering input | CIFAR-10 |
| --- | --- |
| local classifiers | 68.39±0.71 |
| balanced classifiers | 73.21±0.82 |
| oracle | 73.55±0.27 |

Table 4: Ablation study on alignment methods. We run experiments under sudden drift setting and $Dir(0.5)$. We apply partial participation.

| Alignment methods | CIFAR-10 |
| --- | --- |
| w/o alignment | 71.40±0.43 |
| w/o clustered | 72.77±0.71 |
| w/ clustered | 73.21±0.82 |

Table 5: Ablation study on adaptive alignment weight. We apply partial participation. Experiments are run on CIFAR-10 and all results are averaged over 3 runs (mean ± std).

| Weight | $Dir(0.01)$ | $Dir(0.1)$ | $Dir(0.5)$ |
| --- | --- | --- | --- |
| fixed (0) | 62.46±0.92 | 69.33±1.47 | 72.78±0.53 |
| fixed (0.1) | 57.86±1.91 | 70.72±0.83 | 73.83±0.53 |
| fixed (1.0) | 40.89±3.97 | 67.43±1.08 | 74.80±0.41 |
| $\gamma = 10$ | 60.30±0.79 | 69.75±0.19 | 74.97±0.48 |
| $\gamma = 20$ | 61.29±0.59 | 70.21±0.48 | 74.44±0.26 |
| $\gamma = 50$ | 62.15±0.49 | 70.11±1.05 | 73.32±0.06 |

features. In contrast, when training the entire model together, all layers can simultaneously adapt to the training data, allowing them to synergize effectively.

**Performance under distributed concept drift.** Next, we evaluate all methods under the distributed concept drift setting. Table 2 presents the generalized accuracy under this setting. We observe that FedCCFA significantly outperforms all baselines. Specifically, under full participation (i.e., 20 clients with 100% participation), FedCCFA outperforms the second-best method 2.48%, 9.50% and 0.06% on Fashion-MNIST, CIFAR-10 and CINIC-10, respectively; under partial participation (i.e., 100 clients with 20% participation), FedCCFA outperforms the second-best method 1.14%, 8.48% and 3.34% on Fashion-MNIST, CIFAR-10 and CINIC-10, respectively. Compared with the no drift setting, the results of accuracy decrease are shown in Appendix C.2. These results also show that (1) personalized FL methods can effectively adapt to distributed concept drift, demonstrated by the smaller accuracy decrease compared with single-model methods; (2) clustered FL methods may fail to adapt to distributed concept drift under data heterogeneity, because clustering results may be affected by data heterogeneity.

**Effects of balanced classifier training.** As analyzed in Section 4.1, balanced classifier training is crucial for classifier clustering. We conduct experiments under sudden drift setting and Table 3 shows the results of our ablation study on the input of clustering. We see that using the weights of balanced classifiers can significantly increase the generalized accuracy. Furthermore, to demonstrate that accurate clustering results contribute to this improvement, we compare our method against an oracle method which has knowledge about the accurate clustering results at each round. Experimental results show that oracle method indeed increases the generalized accuracy and our method is comparable to the oracle method. These results demonstrate that classifier clustering can enhance the generalization performance and our method achieves accurate clustering.

**Effects of clustered feature anchors.** To verify the efficacy of clustered feature anchors, we ablate FedCCFA by a variety of alignment methods. From Table 4, we see that (1) feature alignment can boost the accuracy under data heterogeneity; (2) clustered feature anchors can facilitate more precise feature alignment under distributed concept drift.

**Effects of adaptive alignment weight.** We turn the degree of data heterogeneity ($\alpha \in \{0.01, 0.1, 0.5\}$ and smaller $\alpha$ corresponds to more severe heterogeneity), and report the average accuracy in Table 5. First, we conduct experiments with various fixed alignment weights $\{0, 0.1, 1.0\}$. Experimental results show that small alignment weight is better under severe data heterogeneity. In addition, under $Dir(0.01)$, training without feature alignment outperforms training with feature alignment, indicating that feature alignment impedes main task learning under this setting. Then, we turn the

scaling factor $\gamma \in \{20, 50, 100\}$. We see that our adaptive alignment weight is robust to the degree of data heterogeneity and the performance is not sensitive to the scaling factor.

## 6  Conclusion

In this work, we take a further step towards distributed concept drift in federated learning. We have shown that FedCCFA can effectively adapt to distributed concept drift, and outperforms existing methods under various concept drift settings. Further experiments suggest that our classifier clustering method significantly enhances the generalization performance for decoupled FL methods. Besides, our clustered feature anchors can enhance the precision of feature alignment and our adaptive alignment weight can stabilize the training process when using feature alignment under severe heterogeneity. We hope that FedCCFA can inspire more works on classifier collaboration and other types of concept drift in federated learning.

**Limitations.**    FedCCFA takes some steps to train balanced classifiers, which increases computation cost. Although we reduce it by setting small iteration and batch size, it is preferable to remove these steps. We will investigate other efficient methods in the future. Besides, feature alignment, used in FedCCFA, FedFM and FedPAC, could impede main task learning under severe data heterogeneity. Therefore, research on the feature alignment under severe heterogeneity is also of interest.

## Acknowledgements

This work was supported by the National Natural Science Foundation of China [No.62172042].

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

# A  Algorithm

The procedure of FedCCFA is formally presented in Algorithm 1. Compared with decoupled FL methods [6, 31], the sections with green color and red color are the additional computation and communication steps, respectively.

**Local training.**  At each round $t$, each selected client $k \in \mathcal{I}^{(t)}$ first updates its extractor $\boldsymbol{\theta}_k^{(t)}$ by the global extractor $\boldsymbol{\theta}^{(t)}$. Then, it starts local training procedure. Lines 6-7, Line 8, Line 9 and Lines 10-12 correspond to balanced classifier training, local classifier training, extractor training and local anchor calculation, respectively. Since the classifier $\boldsymbol{\phi}_k^{(t)}$ learns the conditional distribution $P_k^{(t)}(\mathcal{Y}|\mathcal{X})$, concept drift will not lead to larges gradients in the representation.

**Global aggregation.**  After receiving the local parameters and local anchors, the server starts the aggregation procedure. Line 15, Lines 17-18 and Line 20 correspond to extractor aggregation, class-level clustering and the aggregation of classifiers and anchors, respectively. Note that, for better description, each client updates its classifiers and global anchors in a double loop, as described in Line 21. In the implementation, the server sends aggregated class classifiers and global anchors to each client at the end of round $t$.

---

**Algorithm 1** FedCCFA

---

1: **Input:** number of communication rounds $T$, initial model $\mathbf{w} = \{\boldsymbol{\theta}, \boldsymbol{\phi}\}$, extractor learning rate $\eta_{\boldsymbol{\theta}}$, classifier learning rate $\eta_{\boldsymbol{\phi}}$, number of local epochs $E$, iterations of balanced training $s$, maximum distance $\epsilon$.
2: Initialize $\mathbf{w}^{(0)} = \{\boldsymbol{\theta}^{(0)}, \boldsymbol{\phi}^{(0)}\}$
3: **for** $t = 0, 1, \ldots, T - 1$ **do**
4:      Sample clients $\mathcal{I}^{(t)} \subseteq [K]$ and broadcast $\{\boldsymbol{\theta}^{(t)}, \boldsymbol{\phi}^{(0)}\}$ to each selected client
5:      **for** client $k \in \mathcal{I}^{(t)}$ in parallel **do**
6:          Set $\boldsymbol{\theta}_k^{(t)} = \boldsymbol{\theta}^{(t)}$, $\boldsymbol{\phi}_k^{(t)} = \boldsymbol{\phi}^{(0)}$, and sample a balanced batch $b_k^{(t)}$
7:          Update $\boldsymbol{\phi}_k^{(t)}$ as (3) for $s$ iterations, and save it as $\hat{\boldsymbol{\phi}}_k^{(t)}$     ▷ train balanced classifier
8:          Set $\boldsymbol{\phi}_k^{(t)} = \boldsymbol{\phi}_k^{(t-1)}$, and update $\boldsymbol{\phi}_k^{(t)}$ as (3) for 1 epoch     ▷ train local classifier
9:          Update $\boldsymbol{\theta}_k^{(t)}$ as (9) for $E$ epochs                        ▷ train extractor
10:          **for** class $c \in [C]$ **do**
11:              compute the local feature anchor $a_{k,c}^{(t)}$ as (6)
12:          **end for**
13:          Send $\hat{\boldsymbol{\phi}}_k^{(t)}$, $\boldsymbol{\phi}_k^{(t)}$, $\boldsymbol{\theta}_k^{(t)}$ and $\{a_{k,c}^{(t)}\}_{c=1}^{C}$ to server
14:      **end for**
15:      Update global extractor: $\boldsymbol{\theta}^{(t+1)} = \frac{1}{\sum_{k \in \mathcal{I}^t} |D_k^{(t)}|} \sum_{k \in \mathcal{I}^t} |D_k^{(t)}| \boldsymbol{\theta}_k^{(t)}$
16:      **for** class $c \in [C]$ **do**
17:          $\forall i, j \in \mathcal{I}^{(t)}$: calculate class-level distance $\mathcal{D}_c(i, j)$ between $\hat{\boldsymbol{\phi}}_{i,c}^{(t)}$ and $\hat{\boldsymbol{\phi}}_{j,c}^{(t)}$
18:          $\mathcal{S}_c^{(t)} \leftarrow \text{DBSCAN}(\mathcal{D}_c, \epsilon)$
19:          **for** each cluster $\mathcal{S}_{m,c}^{(t)} \in \mathcal{S}_c^{(t)}$ **do**
20:              get $\bar{\boldsymbol{\phi}}_{m,c}^{(t)}$ and $\mathcal{A}_{m,c}^{(t)}$ as (5) and (7), respectively
21:              client $k$ sets $\boldsymbol{\phi}_{k,c}^{(t)} = \bar{\boldsymbol{\phi}}_{m,c}^{(t)}$ and appends $\mathcal{A}_{m,c}^{(t)}$ to $\mathcal{A}_k^{(t)}$, $\forall k \in \mathcal{S}_{m,c}^{(t)}$
22:          **end for**
23:      **end for**
24: **end for**

---

## B  Experimental details

All experiments are conducted using PyTorch and all results are produced with an Intel Core i7-13700K CPU and a NVIDIA GeForce RTX 4090 GPU. The required memory is around 2GB - 7GB.

### B.1  Datasets and models

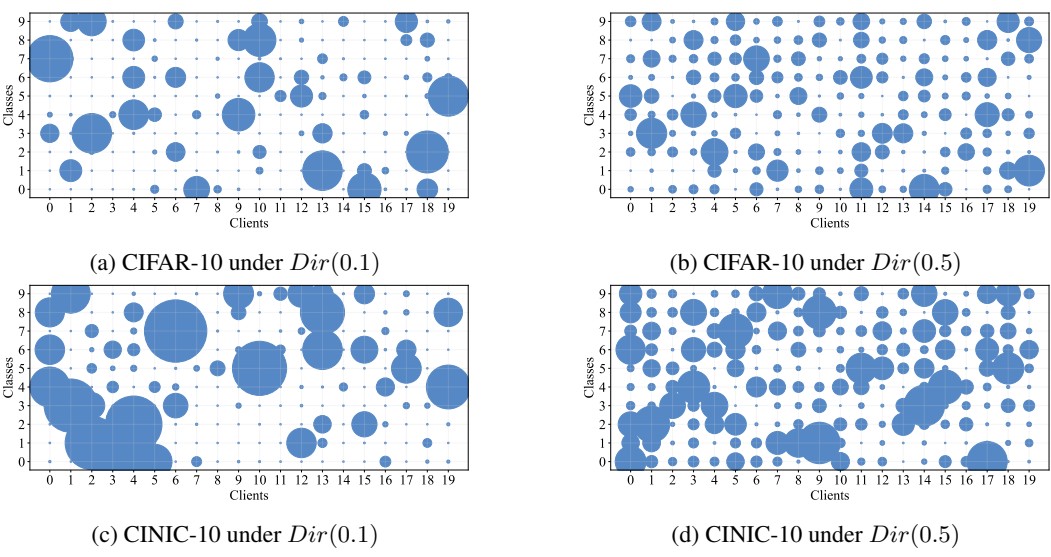

(a) CIFAR-10 under $Dir(0.1)$

(b) CIFAR-10 under $Dir(0.5)$

(c) CINIC-10 under $Dir(0.1)$

(d) CINIC-10 under $Dir(0.5)$

Figure 3: Data partition visualization under 20 clients with full participation.

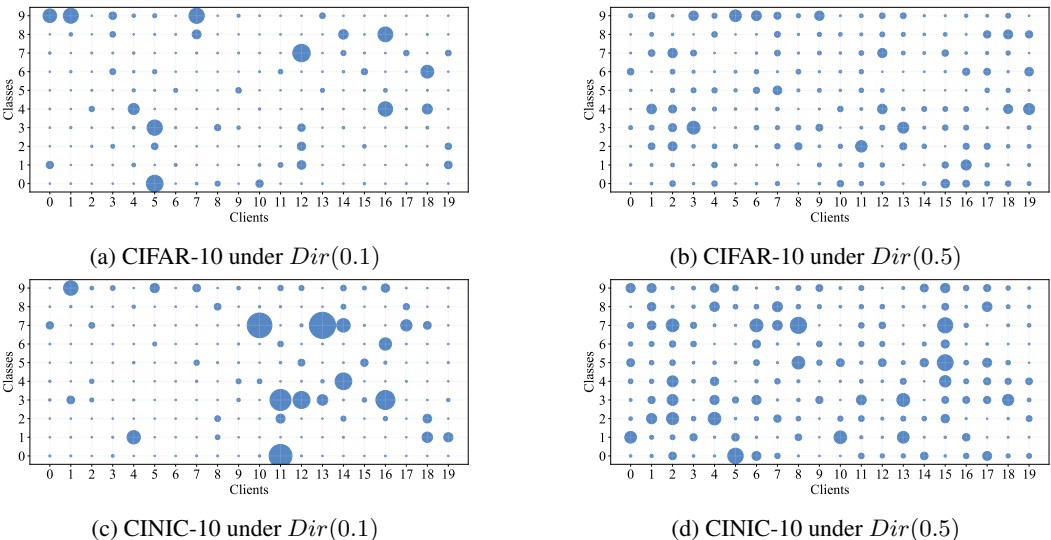

(a) CIFAR-10 under $Dir(0.1)$

(b) CIFAR-10 under $Dir(0.5)$

(c) CINIC-10 under $Dir(0.1)$

(d) CINIC-10 under $Dir(0.5)$

Figure 4: Data partition visualization under 100 clients with 20% participation.

We evaluate our method and all baselines on three datasets: (1) Fashion-MNIST [42]: a dataset comprising of 28×28 grayscale images of 70,000 fashion products from 10 categories (7,000 images per category), where the training set has 60,000 images and the test set has 10,000 images; (2) CIFAR-10 [21]: a dataset consisting of 60,000 32*32 color images in 10 classes (6,000 images per class), where the training set has 50,000 images and the test set has 10,000 images; (3) CINIC-10 [7]: an augmented extension of CIFAR-10, containing the images from CIFAR-10 (60,000 images) and a selection of ImageNet (210,000 images downsampled to 32*32). We use the same model architectures

as those used in FedPAC [43]. For Fashion-MNIST, we construct a CNN model consisting of two convolution layers with 16 5*5 and 32 5*5 filters respectively, each followed by a max pooling layer, and two fully-connected layers with 128 and 10 neurons before softmax layer. For CIFAR-10 and CINIC-10, the CNN model is similar to the model used for Fashion-MNIST but has one more convolution layer with 64 5*5 filters.

We visualize the data partition for CIFAR-10 and CINIC-10 under two degrees of data heterogeneity (i.e., $Dir(0.1)$ and $Dir(0.5)$). Figure 3 shows the partition visualization under full participation (i.e., 20 clients with 100% participation). Since CINIC-10 is much larger than CIFAR-10 and we consider the setting where clients have different data sizes (i.e., quantity skew), data distribution of CINIC-10 is more biased. Figure 4 shows the partition visualization under partial participation (i.e., 100 clients with 20% participation). We randomly choose 20 clients for visualization. We see that data heterogeneity is more severe under full participation, due to more severe quantity skew.

## B.2 Hyperparameters

We employ SGD as the optimizer for all methods. The batch size is set to 64, the momentum is set to 0.9 and the weight decay is set to 0.00001. For all datasets, we set local epochs $E = 5$. In particular, for the decoupled methods, we first train the classifier for 1 epoch with learning rate $\eta_\phi = 0.1$, and then train the extractor for 5 epochs with learning rate $\eta_\theta = 0.01$, aiming to reduce the computation cost; for the other methods, we set local learning rate $\eta = 0.01$.

For FedCCFA, we turn the scaling factor $\gamma \in \{5.0, 10.0, 20.0, 50.0, 100.0\}$, and set it to 20.0 according to the results in Figure 5; we turn the maximum distance $\epsilon \in \{0.05, 0.10, 0.15, 0.20\}$, and set it to 0.10 according to the results in Figure 6.

For IFCA, we set the number of clusters $k = 4$, which is the oracle knowledge of the ground-truth clustering. For the following baselines, we tune their hyperparameters on CIFAR-10 under no concept drift and $Dir(0.5)$, and select the best hyperparameters for all settings:

**FedDrift:** we turn the detection threshold $\delta \in \{0.1, 0.2, 0.5, 1.0, 5.0\}$, and set it to 0.5.

**FedFM:** we turn the penalty coefficient $\lambda \in \{0.1, 0.5, 1.0, 5.0, 10.0, 50.0\}$, and set it to 1.0.

**FedPAC:** we turn the penalty coefficient $\lambda \in \{0.1, 0.5, 1.0, 5.0, 10.0, 50.0\}$, and set it to 1.0.

**Flash:** we turn the server learning rate $\eta_g \in \{0.005, 0.01, 0.02, 0.1, 1.0\}$, and set it to 0.01.

**SCAFFOLD:** we turn the server learning rate $\eta_g \in \{0.005, 0.01, 0.02, 0.1, 1.0\}$, and set it to 1.0.

**pFedMe:** we turn $K \in \{1, 5, 7\}$, and set it to 5; we turn $\lambda \in \{0.1, 0.5, 1.0, 5.0, 10.0, 15.0\}$, and set it to 1.0.

**Ditto:** we turn penalty coefficient $\lambda \in \{0.1, 0.5, 1.0, 5.0\}$, and set it to 5.0.

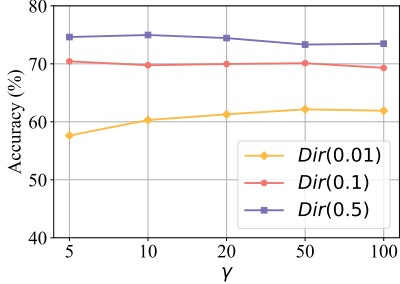

Figure 5: Ablation study on scaling factor $\gamma$. $\gamma = 20$ is an optimal selection.

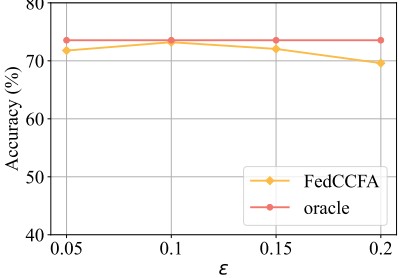

Figure 6: Ablation study on maximum distance $\epsilon$ in DBSCAN. $\epsilon = 0.1$ is an optimal selection.

# C   Additional experimental results

## C.1   Performance under Dir(0.1)

We evaluate all methods under no drift setting and $Dir(0.1)$. Table 6 shows the generalized accuracy under $Dir(0.1)$. We observe that, on CIFAR-10 and CINIC-10 with 20% participation, FedCCFA outperforms baselines. Besides, FedCCFA outperforms all decoupled methods and personalized FL methods, showing that FedCCFA boosts the generalization performance of classifiers under this setting. We find that gradient explosion occurs when adopting FedPAC and FedRep on CINIC-10 with full participation. Under this setting, data heterogeneity is much severe (as shown in Figure 3) and classifier is extremely biased. Such classifier could lead to gradient explosion when using high classifier learning rate, so the accuracy of FedRep may drop to 10% and FedPAC fails to finish FL training due to abnormal inputs to its classifier collaboration. Since FedBABU uses the initial global classifier during training and FedCCFA uses clustered classifiers, this problem is alleviated.

Table 6: Generalized accuracy under $Dir(0.1)$. The sample ratio is 100% and 20% for 20 clients and 100 clients, respectively. All results are averaged over 3 runs (mean ± std). FedPAC fails to finish FL training due to abnormal inputs to its classifier collaboration.

| Method | Fashion-MNIST | | CIFAR-10 | | CINIC-10 | |
|---|---|---|---|---|---|---|
| | 20 clients | 100 clients | 20 clients | 100 clients | 20 clients | 100 clients |
| FedAvg | 88.89±0.30 | **89.01±0.24** | 70.85±0.40 | 67.90±1.03 | 53.17±0.94 | 51.84±0.09 |
| FedProx | 88.75±0.11 | 88.92±0.57 | 71.25±0.63 | 67.48±0.05 | 52.97±0.44 | 52.18±0.83 |
| SCAFFOLD | 86.56±0.25 | 85.52±0.06 | 65.03±0.78 | 51.60±0.41 | 36.24±3.56 | 42.24±1.53 |
| FedFM | 88.43±0.26 | 88.68±0.18 | **71.53±0.37** | 68.31±1.31 | **53.86±0.38** | 50.05±0.92 |
| AdapFedAvg | 88.78±0.25 | 88.91±0.20 | 71.23±0.72 | 66.91±0.65 | 53.15±0.12 | 51.15±0.17 |
| Flash | **89.47±0.27** | 88.76±0.21 | 69.60±0.01 | 68.34±0.48 | 51.53±0.22 | 52.37±0.96 |
| pFedMe | 77.89±0.07 | 83.07±0.18 | 51.35±0.11 | 50.89±0.04 | 34.07±0.21 | 39.89±0.55 |
| Ditto | 87.22±0.02 | 88.00±0.08 | 66.74±0.37 | 65.18±0.04 | 46.49±0.35 | 48.05±0.28 |
| FedRep | 71.17±0.67 | 76.61±0.19 | 44.00±0.30 | 45.09±1.01 | 16.00±10.39 | 30.89±0.60 |
| FedBABU | 70.29±1.36 | 78.39±0.97 | 42.23±0.02 | 48.08±0.86 | 26.34±1.45 | 32.50±0.42 |
| FedPAC | 74.09±0.33 | 84.42±0.11 | 44.48±0.61 | 55.44±0.41 | - | 34.34±0.64 |
| IFCA | 76.52±0.14 | 84.71±0.09 | 49.07±1.20 | 61.33±2.18 | 36.33±1.23 | 43.40±3.39 |
| FedDrift | 88.86±0.02 | 88.88±0.19 | 70.71±0.30 | 66.99±0.57 | 52.34±0.09 | 51.64±0.32 |
| FedCCFA | 88.21±0.40 | 88.32±0.22 | 70.74±0.30 | **70.21±0.48** | 50.13±0.62 | **53.50±0.05** |

## C.2   Performance under sudden drift

We evaluate all methods under distributed concept drift and $Dir(0.1)$. Table 7 presents the generalized accuracy under sudden drift setting. We find that FedCCFA outperforms all baselines on all datasets except for CINIC-10 with full participation. This weakness is caused by wrong clustering results. Specifically, in this case, the performance of global extractor is limited (the accuracy of the best method is only 53.86% under no drift setting), so the poor extractor cannot generate good feature vectors for the classifier. For this reason, the classifier cannot accurately learn the data distribution and its weights may provide wrong information. Besides, we find that gradient explosion also occurs when using FedFM on CINIC-10 with full participation. This is because severe heterogeneity will significantly increase the loss of feature alignment term when distributed concept drift occurs. FL methods without feature alignment are not affected, such as FedAvg, Flash and IFCA.

We present the generalized accuracy decrease under sudden drift setting to show the adaptability to distributed concept drift. These results show that FedCCFA can effectively adapt to distributed concept drift except for CINIC-10 with full participation (due to the poor extractor discussed above). For personalized FL methods, thanks to their personal models or classifiers, distributed concept drift will not affect them.

Table 7: Generalized accuracy under sudden drift and $Dir(0.1)$. The sample ratio is 100% and 20% for 20 clients and 100 clients, respectively. All results are averaged over 3 runs (mean $\pm$ std).

| Method | Fashion-MNIST | | CIFAR-10 | | CINIC-10 | |
|---|---|---|---|---|---|---|
| | 20 clients | 100 clients | 20 clients | 100 clients | 20 clients | 100 clients |
| FedAvg | 67.39±0.72 | 68.00±0.35 | 54.87±0.46 | 53.71±0.95 | 42.15±0.02 | 37.50±0.40 |
| FedProx | 68.43±0.33 | 68.16±0.19 | 55.03±0.19 | 53.57±0.52 | 42.27±0.09 | 38.58±0.22 |
| SCAFFOLD | 65.98±0.76 | 66.30±0.03 | 48.25±1.65 | 41.10±0.64 | 27.78±0.10 | 34.25±0.96 |
| FedFM | 67.89±0.81 | 67.29±0.14 | 55.04±0.87 | 53.01±0.21 | 10.00±0.00 | 36.84±1.21 |
| AdapFedAvg | 66.91±0.74 | 68.07±0.32 | 55.11±0.12 | 52.58±0.54 | **42.43±0.16** | 36.98±0.04 |
| Flash | 67.88±0.13 | 68.96±2.09 | 54.44±0.32 | 54.20±0.59 | 41.35±0.53 | 41.40±0.80 |
| pFedMe | 70.69±0.32 | 71.17±0.07 | 44.54±0.55 | 41.94±0.47 | 30.77±0.08 | 34.54±0.03 |
| Ditto | 72.88±0.33 | 73.29±0.18 | 55.99±0.41 | 52.39±0.14 | 39.60±0.59 | 40.22±0.45 |
| FedRep | 71.49±0.07 | 76.79±0.37 | 43.99±0.27 | 45.81±1.38 | 28.27±0.00 | 31.17±0.04 |
| FedBABU | 67.80±1.75 | 76.93±0.80 | 39.12±1.74 | 46.51±0.20 | 25.80±0.34 | 30.97±0.40 |
| FedPAC | 69.33±0.55 | 79.96±0.42 | 41.35±0.01 | 50.10±0.21 | - | 31.60±0.51 |
| IFCA | 69.45±1.20 | 82.90±0.97 | 43.70±0.21 | 50.34±3.72 | 33.84±2.18 | 31.70±0.30 |
| FedDrift | 63.81±0.02 | 80.48±1.61 | 38.06±3.16 | 46.85±1.07 | 28.32±1.31 | 32.40±2.84 |
| FedCCFA | **85.22±0.08** | **87.63±0.15** | **65.23±0.48** | **66.34±0.96** | 40.94±0.11 | **46.02±0.42** |

Table 8: Generalized accuracy decrease under sudden drift and $Dir(0.5)$. All results are averaged over 3 runs.

| Method | Fashion-MNIST | | CIFAR-10 | | CINIC-10 | |
|---|---|---|---|---|---|---|
| | 20 clients | 100 clients | 20 clients | 100 clients | 20 clients | 100 clients |
| FedAvg | -22.76 | -22.07 | -17.6 | -15.9 | -12.54 | -12.1 |
| FedProx | -21.04 | -21.75 | -17.24 | -15.82 | -12.72 | -12.29 |
| SCAFFOLD | -20.41 | -18.06 | -17.17 | -13.14 | -12.58 | -9.75 |
| FedFM | -21.55 | -21.08 | -18.36 | -17.38 | -12.71 | -12.43 |
| AdapFedAvg | -21.12 | -21.53 | -17.36 | -15.68 | -12.1 | -12.46 |
| Flash | -80.19 | -18.78 | -18.17 | -14.64 | -11.67 | -11.02 |
| pFedMe | -2.69 | -7.15 | -4.16 | -7.34 | -2.41 | -4.04 |
| Ditto | -11.82 | -10.11 | -9.58 | -8.91 | -8.24 | -7.93 |
| FedRep | 0.29 | 0.68 | 0.43 | 1.11 | 0.37 | -0.33 |
| FedBABU | 3 | 0.99 | -0.12 | -0.13 | 0.5 | -0.94 |
| FedPAC | -1.01 | -0.37 | -1.49 | -3.04 | 0.55 | -1.28 |
| IFCA | 0.03 | -0.36 | -2.86 | -15.22 | -4.39 | -12.41 |
| FedDrift | -11.43 | -2.58 | -30.42 | -16.44 | -27.89 | -12.88 |
| FedCCFA | -0.31 | -0.35 | -1.43 | -1.23 | -10.01 | -6.08 |

## C.3 Performance under incremental drift

For all methods, we validate the adaptability to incremental drift, where three different concept drifts occur at round 100, 110 and 120, respectively. Table 9 shows that, under $Dir(0.5)$, FedCCFA can also adapt to this drift setting and outperforms all baselines.

Table 9: Generalized accuracy under incremental drift and $Dir(0.5)$. The sample ratio is 100% and 20% for 20 clients and 100 clients, respectively. All results are averaged over 3 runs (mean $\pm$ std).

| Method | Fashion-MNIST | | CIFAR-10 | | CINIC-10 | |
|---|---|---|---|---|---|---|
| | 20 clients | 100 clients | 20 clients | 100 clients | 20 clients | 100 clients |
| FedAvg | 69.47±1.13 | 68.15±0.90 | 61.31±0.60 | 58.17±0.49 | 49.77±0.33 | 46.33±0.32 |
| FedProx | 70.04±0.70 | 69.06±0.20 | 61.02±0.35 | 57.77±0.43 | 50.04±0.12 | 46.42±0.43 |
| SCAFFOLD | 68.18±0.38 | 69.15±0.44 | 58.14±1.08 | 45.39±0.46 | 46.53±0.28 | 38.06±0.73 |
| FedFM | 68.54±0.38 | 68.86±0.51 | 61.02±0.05 | 57.68±0.48 | 50.86±0.10 | 46.52±0.36 |
| AdapFedAvg | 68.75±0.49 | 68.86±0.39 | 60.74±0.10 | 57.93±0.17 | 49.92±0.32 | 46.00±0.54 |
| Flash | 67.82±0.47 | 70.41±0.12 | 60.61±0.10 | 60.66±0.22 | 49.38±0.06 | 49.77±0.13 |
| pFedMe | 82.38±0.41 | 77.18±0.20 | 58.53±0.43 | 44.01±0.55 | 30.53±14.52 | 37.84±0.81 |
| Ditto | 77.90±0.69 | 79.56±0.21 | 67.57±0.25 | 63.02±0.41 | 50.85±0.63 | 48.61±0.19 |
| FedRep | 82.10±0.14 | 80.85±0.17 | 63.89±0.20 | 51.87±0.59 | 43.80±0.47 | 38.53±0.30 |
| FedBABU | 78.85±1.25 | 81.54±0.46 | 58.63±0.36 | 54.98±0.64 | 41.06±0.87 | 40.04±0.18 |
| FedPAC | 84.10±0.36 | 86.66±0.31 | 67.21±0.60 | 64.78±0.37 | 44.07±0.16 | 46.24±0.25 |
| IFCA | 76.38±0.54 | 88.07±0.29 | 59.74±4.34 | 57.63±0.23 | 45.08±2.60 | 45.51±0.36 |
| FedDrift | 79.62±2.15 | 87.36±0.31 | 46.05±0.07 | 56.32±2.33 | 33.63±0.93 | 45.11±0.35 |
| Ours | **89.49±0.10** | **88.96±0.22** | **76.06±0.39** | **73.42±0.26** | **52.15±0.34** | **53.02±0.46** |

## C.4 Performance under reoccurring drift

To verify whether all methods could recover to the same accuracies once the original data distributions come back, we conduct experiments under the reoccurring drift setting. Table 10 shows that all methods can achieve this goal under reoccurring drift and $Dir(0.5)$.

Table 10: Generalized accuracy under reoccurring drift and $Dir(0.5)$. The sample ratio is 100% and 20% for 20 clients and 100 clients, respectively. All results are averaged over 3 runs (mean $\pm$ std).

| Method | Fashion-MNIST | | CIFAR-10 | | CINIC-10 | |
|---|---|---|---|---|---|---|
| | 20 clients | 100 clients | 20 clients | 100 clients | 20 clients | 100 clients |
| FedAvg | 90.38±0.14 | 90.02±0.16 | 78.37±0.56 | 73.56±0.28 | 62.16±0.18 | 58.66±0.04 |
| FedProx | 90.48±0.08 | 90.15±0.20 | 78.40±0.33 | 74.10±0.64 | 62.40±0.16 | 58.45±0.52 |
| SCAFFOLD | 90.22±0.16 | 87.36±0.36 | 75.23±0.06 | 57.26±0.95 | 58.86±0.12 | 46.64±0.50 |
| FedFM | 90.37±0.06 | 90.21±0.14 | 79.25±0.25 | 73.74±0.23 | 62.56±0.46 | 58.66±0.57 |
| AdapFedAvg | 90.49±0.03 | 90.08±0.13 | 78.36±0.17 | 73.24±0.37 | 62.05±0.44 | 58.72±0.15 |
| Flash | 90.04±0.13 | 89.85±0.25 | 77.59±0.60 | 75.69±0.70 | 61.12±0.32 | 59.72±0.46 |
| pFedMe | 85.43±0.15 | 84.77±0.24 | 62.35±0.08 | 51.11±0.14 | 32.61±15.99 | 41.34±1.05 |
| Ditto | 90.26±0.02 | 89.93±0.16 | 77.31±0.03 | 72.27±0.68 | 58.63±0.43 | 55.83±0.01 |
| FedRep | 33.70±41.05 | 81.29±0.26 | 64.09±0.37 | 52.71±0.45 | 43.64±0.56 | 38.60±0.30 |
| FedBABU | 80.85±1.02 | 81.41±0.48 | 60.11±0.23 | 55.49±0.70 | 41.44±0.43 | 41.33±0.52 |
| FedPAC | 85.26±0.21 | 87.50±0.36 | 67.45±0.32 | 65.89±0.65 | 44.92±0.11 | 47.28±0.57 |
| IFCA | 86.96±0.16 | 88.51±0.76 | 69.82±0.57 | 73.43±0.48 | 51.35±2.19 | 57.90±0.60 |
| FedDrift | 88.29±0.72 | 89.43±0.47 | 68.91±1.81 | 73.21±0.91 | 36.91±0.08 | 56.92±0.84 |
| Ours | 90.31±0.42 | 89.67±0.03 | 78.24±0.49 | 73.70±0.43 | 60.86±0.27 | 59.18±0.25 |

## C.5 Visualization of distance matrix

To demonstrate the effectiveness of our classifier clustering, we conduct experiments on CIFAR-10 under sudden drift setting with full participation, and then visualize the distance matrices (i.e., the input of DBSCAN algorithm) in Figure 7. The left shows the distance matrix of class 1. According to the concept drift setting in Section 5.1, for class 1, clients $\{0, 1, 2, 10, 11, 12\}$ should be grouped and the others should be grouped. The right shows the distance matrix of class 5. For this class, clients $\{6, 7, 8, 9, 16, 17, 18, 19\}$ should be grouped and the others should be grouped. These two matrices demonstrate that our clustering method can effectively measure the distance between each client.

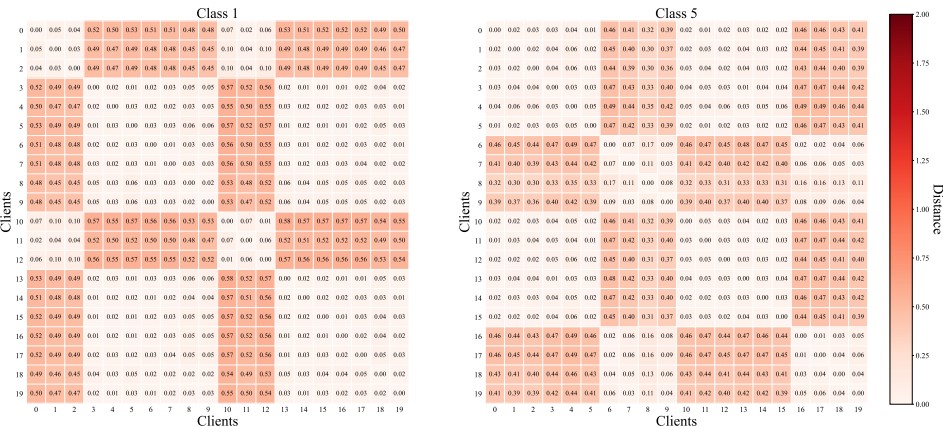

Figure 7: The visualization of distance matrices at round 199 under sudden drift setting.

## C.6 Ablation study on aggregation method

FedCCFA involves class-level classifier aggregation (Equation 5) and class-level feature anchor aggregation (Equation 7). An intuitive manner is sample-number-based weighted aggregation, which is similar to the model aggregation in FedAvg. However, this manner requires clients to upload the sample number for each class, which may leak privacy about clients' category distributions. To address this problem, we consider the uniform aggregation, where all clients share the same aggregation weight. This aggregation manner does not involve other private information. We compare the two manners of aggregation. As shown in Table 11, we observed that uniform aggregation even performs better, especially under concept drift setting. This may be because that uniform aggregation can prevent the classifier from overfitting to the classifier that is significantly biased due to data heterogeneity.

Table 11: Ablation study on aggregation method. We apply partial participation under $Dir(0.5)$.

|  | Fashion-MNIST | CIFAR-10 | CINIC-10 |
| --- | --- | --- | --- |
| no drift (weighted) | 89.39±0.16 | 73.06±0.69 | 57.41±0.12 |
| no drift (uniform) | **89.74±0.18** | **74.44±0.26** | **58.70±0.33** |
| sudden drift (weighted) | 89.14±0.31 | 66.49±1.21 | 45.60±0.38 |
| sudden drift (uniform) | **89.39±0.14** | **73.21±0.82** | **52.62±0.51** |

## C.7 FedDrift under data heterogeneity and distributed concept drift

FedDrift [19] is the first solution for distributed concept drift, which creates new global models based on drift detection and merges models by hierarchical clustering. However, data heterogeneity may disturb model merging, because cluster distance could exceed the threshold under data heterogeneity. To validate this conjecture, we present the accuracy curves and the number of used global models during training process. The latter denotes the number of global models used by all clients at each round. We conduct experiments under the full participation setting.

We first present the accuracy curves and the number of used global models under homogeneous setting in Figure 8. We see that FedDrift can creates new models under various drift settings and accurately merges the models under the same concepts (i.e., the same conditional distributions $P(\mathcal{Y}|\mathcal{X})$). However, since new global models are created at drift rounds and these models need to be retrained, FedDrift cannot rapidly reach a steady state.

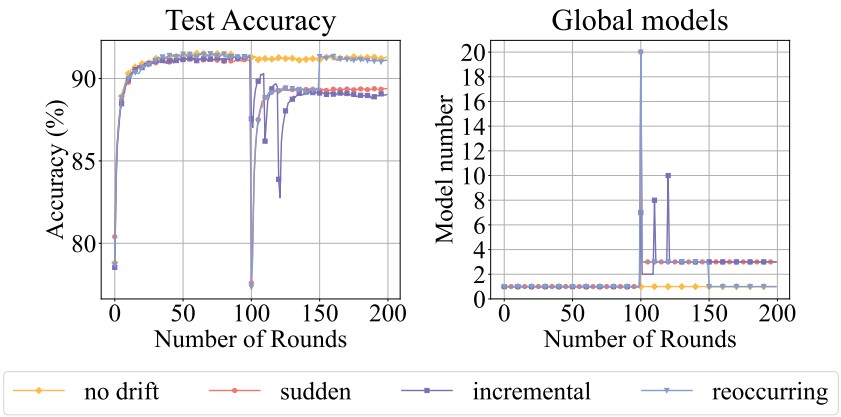

Figure 8: FedDrift on homogeneous Fashion-MNIST.

Then, we present the two indices under $Dir(0.1)$ in Figure 9. We find that FedDrift can creates new model at drift rounds, but it cannot accurately merge the models under the same concepts. This is because, in addition to different concepts, different label distributions can also increase the cluster distances estimated by the loss value, which will prevent model merging. Since all clients choose the only global model at the beginning of training process and the loss delta will not exceed the threshold if no drift occurs, new global models will not be created under the no drift setting.

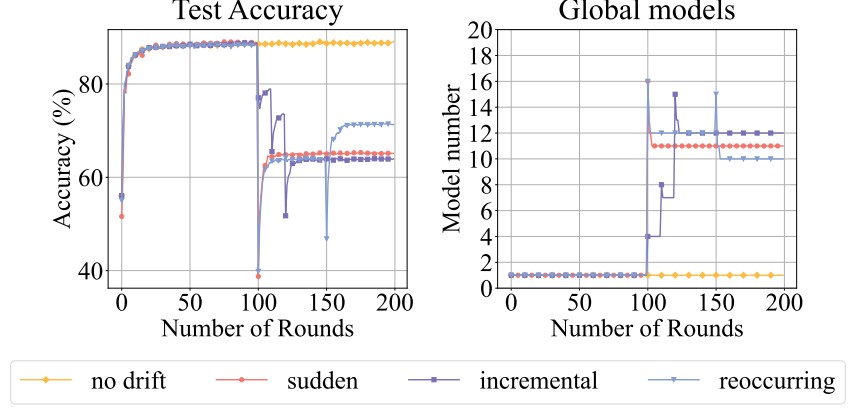

Figure 9: FedDrift on heterogeneous Fashion-MNIST.

## C.8    Computation cost comparison

To enhance client collaboration, FedPAC [43] introduces a classifier combination method, which uses feature statistics to estimate optimal combining weights. However, according to our experimental results, this operation is time-consuming. We compare the computation cost of classifier collaboration between FedCCFA and FedPAC. We count the time consumption for classifier collaboration after sudden drift occurs, and report the average value of 10 rounds. Table 12 shows that FedPAC is more time-consuming than FedCCFA, especially on large datasets (e.g., CINIC-10).

Table 12: Time consumption (seconds) for classifier collaboration. All reults are averaged over 10 rounds.

|  | Fashion-MNIST | CIFAR-10 | CINIC-10 |
|---|---|---|---|
| FedPAC | 1.35 | 1.67 | 2.43 |
| FedCCFA | 0.49 | 0.71 | 0.70 |

## C.9 Personalized performance

Though this work does not focus on personalized FL, we study whether more generalized model could bring benefits to personalization. We distribute test samples by the same Dirichlet distribution used for training data partition, and evaluate the personalized accuracy over each client's personal model (or global extractor followed by personal classifier). After FL training, we fine-tune the classifier with 5 epochs and classifier learning rate is 0.01. Table 13 shows that FedCCFA with classifier fine-tuning is comparable to the most performing personalized methods.

Table 13: Personalized accuracy under no drift setting. FedCCFA-FT denotes FedCCFA with classifier fine-tuning. We apply 20% participation with 100 clients and all experiments are run on CIFAR-10 and results are averaged over 3 runs.

|  | pFedMe | Ditto | FedRep | FedBABU | FedPAC | FedCCFA | FedCCFA-FT |
|---|---|---|---|---|---|---|---|
| $Dir(0.1)$ | 65.94 | 78.84 | 74.26 | 74.46 | 80.24 | 71.99 | 80.52 |
| $Dir(0.5)$ | 60.42 | 79.20 | 68.85 | 71.29 | 79.47 | 75.03 | 79.13 |

