# OpenReview forum: "Classifier Clustering and Feature Alignment for Federated Learning under Distributed Concept Drift"
_NeurIPS.cc/2024/Conference — NeurIPS 2024 poster_

### Official Review · Reviewer_uegQ · 2024-07-10

**Soundness:** 3
**Presentation:** 3
**Contribution:** 2
**Rating:** 5
**Confidence:** 5

**Summary:**

The paper addresses the significant challenge of data heterogeneity and distributed concept drift in federated learning. The authors propose a novel framework, which integrates classifier clustering and feature alignment to improve model performance and collaboration among clients facing different concept drifts.
The key contributions of the paper are as follows:
1. FedCCFA Framework: The proposed framework includes a method for clustering local classifiers at the class level and generating clustered feature anchors to enhance feature alignment. This clustering helps clients with similar data distributions share classifiers, thus improving the generalization performance of the global model.
2. Feature Alignment: The framework introduces an adaptive feature alignment technique that aligns clients' feature spaces based on the entropy of the label distribution. This method helps to alleviate the inconsistencies in feature space due to data heterogeneity.
3. Experimental Validation: Extensive experiments demonstrate that FedCCFA significantly outperforms existing federated learning methods under various concept drift settings. The results show that FedCCFA effectively adapts to distributed concept drift and enhances generalization performance.

**Strengths:**

1. Comprehensive Validation: The research is supported by well-designed experiments and thorough ablation studies, demonstrating the effectiveness of the FedCCFA framework. The mathematical formulations and algorithms are clearly presented, and the results are statistically significant.
2. Clear Presentation: The paper is clearly written and well-structured, making complex concepts accessible. Visual aids, such as graphs and tables, effectively support the explanations. Detailed breakdowns of the experimental setup enhance the reproducibility of the results.
3. Significant Impact: This work addresses a critical gap in federated learning, with potential applications in diverse fields like healthcare, finance, and mobile device collaboration. By improving the adaptability and generalization performance of federated models, the proposed FedCCFA framework provides a robust foundation for future studies to build upon, advancing the current state of federated learning research.

**Weaknesses:**

1. Computation Overhead: The proposed FedCCFA framework involves additional computational steps to train balanced classifiers, which increases the overall computational cost. While the authors attempt to mitigate this by setting small iterations and batch sizes, exploring more efficient methods to achieve balanced classifier training would be beneficial.
2. Limited Evaluation Scenarios: The experiments are primarily conducted on standard datasets. Including more diverse datasets, especially those with real-world distributed concept drift scenarios, would strengthen the validation of the framework's general applicability and robustness.
3. Sensitivity to Hyperparameters: The effectiveness of FedCCFA relies on several hyperparameters, such as the maximum distance in DBSCAN and the scaling factor. While the authors provide some tuning, a more thorough analysis of the sensitivity and robustness to these hyperparameters across different datasets and settings would be valuable.
4. Handling Extreme Data Heterogeneity: The paper addresses data heterogeneity, but extreme cases of data heterogeneity can still pose challenges, as noted with gradient explosions in some scenarios. Further discussion and potential solutions to handle such extreme cases more effectively would improve the robustness of the framework.

**Questions:**

1. Computational Overhead
Have you explored alternative methods for balanced classifier training to reduce computational overhead? It’s better to consider leveraging advanced optimization techniques or lightweight pre-processing steps to achieve balanced classifier training more efficiently.
2. Evaluation on Diverse Datasets
Do you plan to evaluate FedCCFA on more diverse, real-world datasets reflecting practical distributed concept drift scenarios? It’s better to extend evaluation to real-world datasets would provide a comprehensive understanding of the framework's applicability and robustness.
3. Sensitivity Analysis of Hyperparameters
Have you conducted a thorough sensitivity analysis of key hyperparameters across different datasets and settings? A detailed sensitivity analysis could help understand the impact of these hyperparameters on performance and provide guidelines for their selection.
4. Handling Extreme Data Heterogeneity
What solutions have you considered for effectively handling extreme cases of data heterogeneity and preventing gradient explosions? It’s better to investigate adaptive methods that dynamically adjust based on real-time training stability monitoring to handle extreme data heterogeneity more robustly

**Limitations:**

1. Adaptability to Various Drift Patterns: The paper evaluates performance under specific concept drift scenarios but may not cover all possible drift patterns. Expanding the evaluation to include a wider variety of drift patterns would provide a more comprehensive understanding of the framework’s adaptability.
2. Real-World Implementation: The paper demonstrates effectiveness in controlled experimental settings. Discussing potential challenges and solutions for deploying FedCCFA in real-world environments, including scalability and communication efficiency, would strengthen the paper.
3. Detailed Analysis of Classifier Clustering: The paper proposes class-level classifier clustering but provides limited analysis on the clustering's dynamics and potential pitfalls. A deeper analysis of how clustering decisions impact overall model performance and stability would be beneficial.

---

> ### Author Rebuttal · Authors · 2024-08-05
>
> Thank you for your detailed review and thoughtful comments. Below we address specific questions:
>
> > W1 & Q1: Computation Overhead: The proposed FedCCFA framework involves additional computational steps to train balanced classifiers [...] exploring more efficient methods to achieve balanced classifier training would be beneficial.
>
> A1: Thank you for your insightful comments. We appreciate your suggestion to consider more efficient alternatives. In our current work, balanced classifier is trained to prevent classifier bias from misleading client clustering. It is preferable to address classifier bias during local training, such as CCVR [1], FedETF [2] and FedBR [3]. However, (1) CCVR requires virtual representations sampled from an approximated gaussian mixture model and re-trains classifier using these representations; (2) FedETF utilizes a fixed ETF classifier to address classifier bias, which requires finetuning and is not suitable for concept drift setting; (3) FedBR generates pseudo-data to reduce bias in local classifiers, which also requires additional computation cost. We agree that future work could explore more efficient methods, and we plan to investigate this in subsequent studies. For example, some global information (e.g., global feature anchors) can be used to calibrate the local classifier.
>
> > W2 & Q2: Limited Evaluation Scenarios: The experiments are primarily conducted on standard datasets. Including more diverse datasets [...] would strengthen the validation of the framework's general applicability and robustness.
>
> A2: Thank you for your constructive comments. We agree that testing on real-world data is crucial for demonstrating the practical applicability of our method. Unfortunately, there are currently no datasets that fit the specific requirements of our study, since publicly available real-world datasets usually assume that $P(\mathcal{Y}|\mathcal{X})$ is invariant. Therefore, existing works focusing on concept drift in federated learning (e.g., FedDrift [4] and Flash [5]) use synthetic datasets. Note that FMoW dataset used in FedDrift does not contain the changes in $P(\mathcal{Y}|\mathcal{X})$, as described in Section 2.1 of the original paper.
>
> > W3 & Q3: Sensitivity to Hyperparameters: The effectiveness of FedCCFA relies on several hyperparameters, [...] a more thorough analysis of the sensitivity and robustness to these hyperparameters across different datasets and settings would be valuable.
>
> A3: Thank you for your valuable comments. We have conducted additional experiments to thoroughly analyze the sensitivity of key hyperparameters across different datasets and concept drift settings. Experimental results show that $\gamma$ ranging from 10 to 50 tends to perform better, and $\epsilon \le 0.1$ is better.
>
> | | $\gamma$ = 5 | 10 | 20 | 50 | 100 |
> |---|---|---|---|---|---|
> |Fashion-MNIST (no drift) | 89.84$\pm$0.11 | 89.63$\pm$0.23 | 89.74$\pm$0.18 | 89.70$\pm$0.19 | 89.37$\pm$0.07 |
> |Fashion-MNIST (sudden drift) | 89.73$\pm$0.32 | 89.00$\pm$0.19 | 89.39$\pm$0.14 | 89.01$\pm$0.17 | 89.15$\pm$0.22 |
> |CIFAR-10 (no drift) | 75.19$\pm$0.15 | 74.97$\pm$0.48 | 74.44$\pm$0.26 | 73.32$\pm$0.06 | 73.47$\pm$0.41 |
> |CIFAR-10 (sudden drift) | 70.94$\pm$0.42 | 73.17$\pm$0.77 | 73.21$\pm$0.82 | 72.25$\pm$0.32 | 71.45$\pm$0.68 |
>
> | | $\epsilon$ = 0.05 | 0.1 | 0.15 | 0.2 |
> |---|---|---|---|---|
> |Fashion-MNIST (no drift) | 89.13$\pm$0.29 | 89.74$\pm$0.18 | 89.53$\pm$0.10 | 89.76$\pm$0.16 |
> |Fashion-MNIST (sudden drift) | 88.29$\pm$0.41 | 89.39$\pm$0.14 | 89.13$\pm$0.34 | 89.37$\pm$0.15 |
> |CIFAR-10 (no drift) | 74.00$\pm$0.18 | 74.44$\pm$0.26 | 74.18$\pm$0.13 | 73.28$\pm$0.69 |
> |CIFAR-10 (sudden drift) | 72.99$\pm$0.78 | 73.21$\pm$0.82 | 71.40$\pm$0.84 | 69.33$\pm$0.50 |
>
> > W4 & Q4: Handling Extreme Data Heterogeneity: The paper addresses data heterogeneity, but extreme cases of data heterogeneity [...] Further discussion and potential solutions to handle such extreme cases more effectively would improve the robustness of the framework.
>
> A4: Thank you for your insightful comments. In FL, different clients may suffer from various level of data heterogeneity. Requiring all clients to use the same hyperparameters for local training is not suitable. To address this problem, we propose adaptive alignment weight. We agree that other adaptive methods that dynamically adjust based on real-time training stability monitoring are more helpful to address this problem. An intuition is gradient clipping. According to our observations, gradient clipping can effectively address the gradient explosions when using FedFM, FedPAC, etc. Since the main focus in this work is distributed concept drift adaptation, we will investigate other adaptive methods to improve the performance under severe heterogeneity.
>
> **References:**
>
> [1] Luo, M., Chen, F., Hu, D., Zhang, Y., Liang, J., & Feng, J. (2021). No fear of heterogeneity: Classifier calibration for federated learning with non-iid data. Advances in Neural Information Processing Systems, 34, 5972-5984.
>
> [2] Li, Z., Shang, X., He, R., Lin, T., & Wu, C. (2023). No fear of classifier biases: Neural collapse inspired federated learning with synthetic and fixed classifier. In Proceedings of the IEEE/CVF International Conference on Computer Vision (pp. 5319-5329).
>
> [3] Guo, Y., Tang, X., & Lin, T. (2023, July). Fedbr: Improving federated learning on heterogeneous data via local learning bias reduction. In International Conference on Machine Learning (pp. 12034-12054). PMLR.
>
> [4] Jothimurugesan, E., Hsieh, K., Wang, J., Joshi, G., & Gibbons, P. B. (2023, April). Federated learning under distributed concept drift. In International Conference on Artificial Intelligence and Statistics (pp. 5834-5853). PMLR.
>
> [5] Panchal, K., Choudhary, S., Mitra, S., Mukherjee, K., Sarkhel, S., Mitra, S., & Guan, H. (2023, July). Flash: concept drift adaptation in federated learning. In International Conference on Machine Learning (pp. 26931-26962). PMLR.

---

> > ### Comment · Reviewer_uegQ · 2024-08-12
> >
> > Thank you for your responses, and I will raise my score.

---

### Official Review · Reviewer_EQ7w · 2024-07-11

**Soundness:** 3
**Presentation:** 3
**Contribution:** 3
**Rating:** 6
**Confidence:** 4

**Summary:**

This paper explores the impact of distributed concept drift on federated learning (FL), and proposes a novel FL framework, FedCCFA, to adapt to distributed concept drift with data heterogeneity (i.e., label distribution shift). Extensive experiments demonstrated the effectiveness and generality of the proposed method.

**Strengths:**

1. Federated learning (FL) under concept drift holds significant academic and practical value. The method proposed in this paper is both reasonable and effective.
2. The experiments are extensive, not only comparing the proposed method with SOTA methods but also simulating various types of drift. This demonstrates the method's generality across different drift scenarios.
3. The paper is well-organized and well-written, enhancing the clarity and impact of its findings.

**Weaknesses:**

1. In the introduction, the authors use the distribution of medical images as an example to describe data heterogeneity. However, the actual task addressed in the paper is label distribution shift, which does not align with the previous example. It is recommended that the authors directly use label distribution shift to describe the problem addressed in the paper
2. The authors calculate the global feature anchor by averaging (i.e., Eq. (7)). However, given the presence of distributed concept drift among different clients, simply averaging may be unreasonable. It raises the question of whether concept drift adaptation should be considered.
3. The adaptive alignment weight is calculated based on the entropy of the label distribution. However, there lacks the relevant motivation, references, and theoretical justification for this strategy.
4. Federated learning (FL) under distributed concept drift primarily addresses concept drift across clients and over time. This is similar to the research problem of multistream classification under concept drift [1-3]. It is recommended that the authors discuss these works to enhance readers' understanding of related research.
    [1] An adaptive framework for multistream classification, CIKM 2016
    [2] Fusion: An online method for multistream classification, CIKM 2017
    [3] Online boosting adaptive learning under concept drift for multistream classification, AAAI 2024
5. The paper does not provide open-source code and datasets, which compromises the reproducibility of the experiments.

**Questions:**

discussed in weaknesses.

**Limitations:**

yes

---

> ### Author Rebuttal · Authors · 2024-08-05
>
> Thank you for your constructive comments. Below we address specific questions.
>
> > W1: In the introduction, the authors use the distribution of medical images [...] It is recommended that the authors directly use label distribution shift to describe the problem addressed in the paper.
>
> A1: We believe there may have been a misunderstanding regarding the primary problem our research tackles. To clarify, the primary problem our paper addresses is distributed concept drift with data heterogeneity, not just label distribution shift. We guess that the "label distribution shift" you mentioned is "data heterogeneity" in our paper, where the marginal distribution $P(\mathcal{Y})$ varies across clients. In our paper, we consider a new problem named distributed concept drift. Specifically, the conditional distribution $P(\mathcal{Y}|\mathcal{X})$ may vary across clients (e.g., diagnoses can vary among doctors) and change over time (e.g., a doctor may offer different diagnoses at different times). These two kinds of changes constitute distributed concept drift. As for data heterogeneity, we think it is a common setting for federated learning, so we conduct experiments under two Dirichlet distribution (Dir(0.1) and Dir(0.5)) to simulate data heterogeneity.
>
> > W2: The authors calculate the global feature anchor by averaging (i.e., Eq. (7)). [...] It raises the question of whether concept drift adaptation should be considered.
>
> A2: Thank you for your valuable feedback. We would like to clarify that this issue has already been addressed in our manuscript. As detailed in "Clustered feature anchors" in Section 4.2, given the presence of distributed concept drift, simply averaging clients' local anchors could generate incorrect global anchors, which can mislead feature alignment. To address this problem, we propose clustered feature anchors (i.e., $\mathcal{A}\_{m, c}^{(t)}$ in Eq. (7)), where the local anchors are averaged "for each cluster $\mathcal{S}\_{m, c}^{(t)}$". Since the conditional distribution $P(\mathcal{Y}|\mathcal{X})$ of clients in the same cluster is similar, the "clustered feature anchors" can correctly guide the feature alignment. The experimental results in Table 4 demonstrate the efficacy of our clustered feature anchors.
>
> > W3: The adaptive alignment weight is calculated based on the entropy of the label distribution. However, there lacks the relevant motivation, references, and theoretical justification for this strategy.
>
> A3: Thank you for your constructive comments. In our ablation study on alignment weight (Table 5), we observed that alignment weight should be reduced under severe heterogeneity. Then, we propose adaptive alignment weight based on the intuition that the degree of data heterogeneity can be reflected by the entropy of marginal distribution $P(\mathcal{Y})$. This motivation can be supported by Shannon's information theory [1]. Besides, He and Garcia [2] also note that imbalanced distributions result in lower entropy because the majority class dominates, reducing the unpredictability of the outcomes. We will clarify this part of our manuscript. We acknowledge the importance of providing a thorough theoretical analysis for our adaptive alignment weight. However, our current work primarily aims to address the feature alignment under distributed concept drift. Further theoretical analysis and other adaptive method will be considered in subsequent studies.
>
> > W4: Federated learning (FL) under distributed concept drift [...] This is similar to the research problem of multistream classification under concept drift [1-3]. It is recommended that the authors discuss these works [...].
>
> A4: We appreciate your pointer to these related works [3-5]. These works focus on multistream classification, which involves two independent non-stationary data generating processes (i.e., source stream and target stream). A sampling bias may exist between the distributions represented by these two streams of data. That is, training and test data distributions are not guaranteed to be similar. Different from multistream classification, distributed concept drift focus on the changing conditional distribution $P(\mathcal{Y}|\mathcal{X})$ across clients and over time. For each client at any round, training and test data distributions are assumed to be similar. We will cite these works and discuss the relationship in our final version, as you suggested.
>
> > W5: The paper does not provide open-source code and datasets, which compromises the reproducibility of the experiments.
>
> A5: Thank you for your valuable feedback. According to the FAQ of NeurIPS 2024, it is not allowed to add any link in any part of the rebuttal. We will provide our code once this paper is accepted.
>
>
> **References:**
>
> [1] Shannon, C. E. (1948). A mathematical theory of communication. The Bell system technical journal, 27(3), 379-423.
>
> [2] He, H., & Garcia, E. A. (2009). Learning from imbalanced data. IEEE Transactions on knowledge and data engineering, 21(9), 1263-1284.
>
> [3] Chandra, S., Haque, A., Khan, L., & Aggarwal, C. (2016, October). An adaptive framework for multistream classification. In Proceedings of the 25th ACM international on conference on information and knowledge management (pp. 1181-1190).
>
> [4] Haque, A., Wang, Z., Chandra, S., Dong, B., Khan, L., & Hamlen, K. W. (2017, November). Fusion: An online method for multistream classification. In Proceedings of the 2017 ACM on Conference on Information and Knowledge Management (pp. 919-928).
>
> [5] Yu, E., Lu, J., Zhang, B., & Zhang, G. (2024, March). Online boosting adaptive learning under concept drift for multistream classification. In Proceedings of the AAAI Conference on Artificial Intelligence (Vol. 38, No. 15, pp. 16522-16530).

---

> > ### Comment · Reviewer_EQ7w · 2024-08-12
> >
> > Thank you for the authors' response. All of my concerns have been addressed, and I will increase the score to 6.

---

### Official Review · Reviewer_erp1 · 2024-07-12

**Soundness:** 3
**Presentation:** 2
**Contribution:** 3
**Rating:** 6
**Confidence:** 3

**Summary:**

The paper proposes a federated learning framework called FedCCFA, which addresses the challenges posed by distributed concept drift and data heterogeneity. The authors introduce innovative solutions such as classifier clustering and adaptive feature alignment to enhance collaboration and improve model performance under various concept drift settings. The authors demonstrate the effectiveness of their approach through experimental results showing significant improvements over existing methods.

**Strengths:**

1. The paper tackles an important and relatively unexplored problem in federated learning, namely concept drift and data heterogeneity.
2. The extensive experimental results demonstrate the effectiveness of FedCCFA, showing significant performance improvements over existing methods in various concept drift scenarios.

**Weaknesses:**

1. I am unable to fully understand the motivation of the paper. Figure 1 does not clearly illustrate how Decoupled Clustering addresses the real drift problem compared to previous decoupled methods. While it shows better performance than FedAvg and pure Decoupled methods, the underlying reasons remain unclear. Why does introducing clustering improve performance?
2. The writing of the paper can be improved. For example, the specific meaning of \(\phi_{i,c}\) is not explained, leading to significant confusion in understanding equation (4). The authors need to explain each symbol to enhance the readability of the paper. Additionally, the term "balanced batch" in line 160 needs to be explained, specifically how it was sampled.

**Questions:**

1. In the experiments, which classifiers were ultimately clustered in the class-level clustering? What does the final distance matrix look like?
2. Equation (5) uses uniform aggregation to avoid privacy concerns. Why does this method avoid privacy concerns? Could taking the average directly lead to performance loss in classifiers?

**Limitations:**

The authors point out that FedCCFA requires training balanced classifiers, which will increase computational complexity to some extent.

---

> ### Author Rebuttal · Authors · 2024-08-05
>
> Thank you for your valuable comments and kind words to our work. Below we address specific questions:
>
> > W1: I am unable to fully understand the motivation of the paper. Figure 1 does not clearly illustrate how Decoupled Clustering [...] the underlying reasons remain unclear. Why does introducing clustering improve performance?
>
> A1: Thank you for your constructive feedback. We will clarify this part of our manuscript. When real drift occurs, decoupled methods can adapt to this drift by training the classifier while fixing the extractor. Each local classifier learns corresponding conditional distribution $P(\mathcal{Y}|\mathcal{X})$. However, some clients may share similar $P(\mathcal{Y}|\mathcal{X})$, and pure decoupled methods neglect the fine-grained collaboration between local classifiers. This will make each client's local classifier overfit to its local data. To address this problem, we propose a novel class-level classifier clustering method, which aggregates the class classifiers within the same cluster. This aggregation reduces the bias introduced by any single client's data, contributing to improved generalization performance. That is, individual local classifiers trained on local data exhibit biased parameter estimation, while the aggregation of multiple classifiers mitigates these biases and improves the overall parameter estimation.
>
> > W2: The writing of the paper can be improved. For example, the specific meaning of $\boldsymbol{\phi}\_{i,c}$ is not explained, [...] Additionally, the term "balanced batch" in line 160 needs to be explained, specifically how it was sampled.
>
> A2: Thank you for your valuable feedback, and we will incorporate the suggestions into our updated draft.
>
> 1. The meaning of $\{\hat{\boldsymbol{\phi}}\_{k, c}^{(t)}\}$ : At round $t$, the server separates each client $k$'s balanced classifier $\hat{\boldsymbol{\phi}}\_k^{(t)}$ for each class $c$ ( referred to as class classifier $\hat{\boldsymbol{\phi}}\_{k, c}^{(t)}$ ). Then, for each class $c$, the server measures the class-level distances $\mathcal{D}\_c(i, j)$ between clients $i$ and $j$ by their class classifiers $\hat{\boldsymbol{\phi}}\_{i, c}^{(t)}$ and $\hat{\boldsymbol{\phi}}\_{j, c}^{(t)}$.
>
> 2. The meaning of "balanced batch": To prevent class imbalance from misleading classifier clustering, FedCCFA uses a balanced batch $b\_k^{(t)}$ to train a balanced classifier $\hat{\boldsymbol{\phi}}\_k^{(t)}$ for distance estimation. To reduce computation cost, we randomly and uniformly select 5 samples for each class $c \in [C]$, i.e., the balanced batch size $|b\_k^{(t)}|$ is $5 \* C$.
>
> > Q1: In the experiments, which classifiers were ultimately clustered in the class-level clustering? What does the final distance matrix look like?
>
> A3: In our experiments, for each class $c \in [C]$, all selected clients' class classifiers $\{\hat{\boldsymbol{\phi}}\_{k, c}^{(t)}\}$ are used for class-level clustering, where $k \in \mathcal{I}^{(t)}$. That is, for each class $c$, the server computes a $|\mathcal{I}^{(t)}| \* |\mathcal{I}^{(t)}|$ distance matrix $\mathcal{D}\_c$ to measure the distance between each client's class classifier. Please see the attached PDF in the top-level comment for the visualization of distance matrix.
>
> > Q2: Equation (5) uses uniform aggregation to avoid privacy concerns. Why does this method avoid privacy concerns? Could taking the average directly lead to performance loss in classifiers?
>
> A4: FedCCFA involves class-level classifier aggregation (Eq. (5)) and class-level feature anchor aggregation (Eq. (7)). An intuitive manner is sample-number-based weighted aggregation, which is similar to the model aggregation in FedAvg. However, this manner requires clients to upload the sample number for each class, which may leak privacy about clients' category distributions. To address this problem, we consider the uniform aggregation, where all clients share the same aggregation weight. This aggregation manner does not involve other private information.
> We compare the two manners of aggregation. Interestingly, we observed that uniform aggregation even performs better, especially under concept drift setting. This may be because that uniform aggregation can prevent the classifier from overfitting to the classifier that is significantly biased due to data heterogeneity.
>
> |  | Fashion-MNIST | CIFAR10 | CINIC-10 |
> | ------------- | -------- | -------- | ----|
> | no drift (weighted) | 89.39$\pm$0.16 | 73.06$\pm$0.69 | 57.41$\pm$0.12 |
> | no drift (uniform) | **89.74$\pm$0.18** | **74.44$\pm$0.26** | **58.70$\pm$0.33** |
> | sudden drift (weighted) | 89.14$\pm$0.31 | 66.49$\pm$1.21 | 45.60$\pm$0.38 |
> | sudden drift (uniform) | **89.39$\pm$0.14** | **73.21$\pm$0.82** | **52.62$\pm$0.51** |
>
> Thank you for your valuable comments. We will incorporate these results into our final version.

---

> > ### Comment · Reviewer_erp1 · 2024-08-12
> >
> > Thanks for the authors' detailed response and additional experiments. My concerns have been adequately addressed. I will keep my positive score.

---

### Official Review · Reviewer_HDwb · 2024-07-12

**Soundness:** 3
**Presentation:** 3
**Contribution:** 3
**Rating:** 7
**Confidence:** 4

**Summary:**

This paper proposes to consider two cross-coupled important issues, i.e., federated learning and concept drift. To address such a challenging problem, the FedCCFA framework has been proposed composed of classifier clustering and feature alignment modules. The former is designed to cope with concept drift and enhance the generalization performance of the model. The latter is utilized to prevent data heterogeneity of clients. The overall design is sound, interesting, and intuitive. The whole paper is well-organized and well-written. The experimental evaluation is comprehensive and convincing.

**Strengths:**

- A relatively new and important problem of federated learning under concept drift is addressed.
- Contributions of this work are in the framework perspective and the module/strategy design perspective.
- Both classifier clustering and feature alignment are sound and interesting ideas.
- The whole paper is well-organized.
- Comprehensive evaluation comparing many counterparts (including SOTAs) on benchmark datasets.

**Weaknesses:**

- Lack of a framework pipeline to intuitively explain the overall design.
- Time complexity analysis is missing.
- Visual classification results on the datasets are preferable.
- The source code is not opened.

**Questions:**

See the weaknesses.

**Limitations:**

Yes

---

> ### Author Rebuttal · Authors · 2024-08-05
>
> Thank you for your valuable comments and kind words to our work. Below we address specific questions:
>
> > W1: Lack of a framework pipeline to intuitively explain the overall design.
>
> A1: Thank you for your valuable feedback. Please see the attached PDF in the top-level comment for our framework pipeline. We will incorporate this pipeline into our final version.
>
> > W2: Time complexity analysis is missing.
>
> A2: Thank you for your constructive comment. We conducted a comprehensive time complexity analysis at client side and compared our FedCCFA to other methods, including FedFM [1], FedRep [2] and FedPAC [3]. $E$ and $B$ denotes the epoch and batch num, respectively. In our FedCCFA, we train a balanced classifier for $s$ iteration with a batch of data. $m$, $f$ and $c$ denotes the computation cost for training local model, extractor and classifier, respectively. Note that, to reduce computation cost, $s$ is small.
>
> | Method | Complexity |
> | --------- | ------------- |
> | FedFM | O(E\*B\*m) |
> | FedRep | O(E\*B\*f+B\*c) |
> | FedPAC | O(E\*B\*f+B\*c) |
> | FedCCFA | O(E\*B\*f+B\*c+s\*c) |
>
> In particular, at server side, our classifier collaboration method is more efficient than that of FedPAC, which can be demonstrated by the results in Appendix B.6.
>
> > W3: Visual classification results on the datasets are preferable.
>
> A3: Thank you for your valuable feedback. Please see the attached PDF in the top-level comment for the visualization of distance matrix. We think it can be helpful to understand our clustering method.
>
> > W4: The source code is not opened.
>
> A4: Thank you for your valuable feedback. According to the FAQ of NeurIPS 2024, it is not allowed to add any link in any part of the rebuttal. We will provide our code once this paper is accepted.
>
>
> **References:**
>
> [1] Ye, R., Ni, Z., Xu, C., Wang, J., Chen, S., & Eldar, Y. C. (2023). Fedfm: Anchor-based feature matching for data heterogeneity in federated learning. IEEE Transactions on Signal Processing.
>
> [2] Collins, L., Hassani, H., Mokhtari, A., & Shakkottai, S. (2021). Exploiting shared representations for personalized federated learning. In International conference on machine learning (pp. 2089-2099). PMLR.
>
> [3] Xu, J., Tong, X., & Huang, S. L. Personalized Federated Learning with Feature Alignment and Classifier Collaboration. In The Eleventh International Conference on Learning Representations.

---

> > ### Comment · Reviewer_HDwb · 2024-08-08
> >
> > Thank you for your responses, I will stick my score with a higher confidence.

---

### Official Review · Reviewer_U5Wz · 2024-07-15

**Soundness:** 3
**Presentation:** 2
**Contribution:** 2
**Rating:** 6
**Confidence:** 4

**Summary:**

The paper introduces a solutions based on clustering to group classifiers in the federated learning setting. Such an approach is meant to deal with concept drift in data so as to track and adapt the evolutions of the classifiers of the clients in federated learning. On of the core points is to align features and this is carried out by ether adaptive feature alignment procedure suggested in the paper.

**Strengths:**

- The idea of letting the global/local models evolve over time is interesting and relevant
- The clustering is based on the classifier space (the feature extraction phase is fixed)
- Results in the concept drift are promising

**Weaknesses:**

Some technical points of the paper are not adequately addressed such as:
- the partitioning of feature extractor + classifier is relevant and requires to share a common feature extractor. It is not clear to this reviewer whether the feature extractor is fixed or adapted over time
- clustering of models requires to assume that the distribution of the model parameters shares some "locality" (e.g., this holds for linear models but not in general for NNs).
- performance in stationary-conditions are less relevant. This is a crucial point (e.g., not suffered by other solutions) that requires to be further investigated

**Questions:**

- What's the need to partition the models into feature extractor and classifier?
- Which is the assumption guaranteeing the clustering of classifier parameters in the parameter space?
- What's the reason of the reduction in accuracy in no-change condition (i.e., Table 1) and how to mitigate this point?

**Limitations:**

see previous box

---

> ### Author Rebuttal · Authors · 2024-08-05
>
> Thank you for your valuable comments and kind words to our work. Below we address specific questions:
>
> > W1: the partitioning of feature extractor + classifier is relevant and requires to share a common feature extractor. It is not clear to this reviewer whether the feature extractor is fixed or adapted over time.
>
> A1: We will clarify this part of our manuscript. As we mentioned in Section 4.3 (FedCCFA), the feature extractor is trained locally by each selected client (line 9 in Algorithm 1) and the server aggregates these clients' local extractors (line 15 in Algorithm 1). The aggregated extractor is broadcasted to the clients at the next round (line 4 in Algorithm 1). The training and aggregation of feature extractors can further enhance generalization performance. In particular, to mitigate the impact of concept drift on the model training, local classifier training (line 8) is conducted before local extractor training, which can effectively adapt to real drift (i.e., the conditional distribution $P(\mathcal{Y}|\mathcal{X})$ changes).
>
> > W2: clustering of models requires to assume that the distribution of the model parameters shares some "locality" (e.g., this holds for linear models but not in general for NNs).
>
> A2: We believe there may have been a misunderstanding regarding our clustering method. Our clustering method is also suitable for NNs. Specifically, in our FedCCFA framework and other decoupled methods (e.g., FedBABU [1] and FedPAC [2]), the classifier is the last fully-connected layer (i.e., a simple linear classifier) and the feature extractor is composed of the remaining layers. The extractor is shared across all clients and maps the input to a low-dimensional feature vector. Once concept drift occurs (i.e., the conditional distribution $P(\mathcal{Y}|\mathcal{X})$ changes), different linear classifiers will be learned and the parameters of these classifiers reflect corresponding $P(\mathcal{Y}|\mathcal{X})$. Finally, the server separates these classifiers for each class (referred to as class classifier) and measures the class-level distances between each client's class classifier, which can group the clients according to their $P(\mathcal{Y}|\mathcal{X})$. The class classifier is still a linear model that outputs the logit for the specific class. Therefore, our method is also suitable for NNs, as long as the last fully-connected layer is used as the local classifier. Our experiments conducted on two CNNs also demonstrate this feasibility. To avoid further confusion, we will clarify this section of our manuscript.
>
> > W3: performance in stationary-conditions are less relevant. This is a crucial point (e.g., not suffered by other solutions) that requires to be further investigated.
>
> A3: The lower accuracy under no drift setting is attributed to decoupled training (i.e., training the classifier first and then training the feature extractor). Specifically, for decoupled methods (e.g., FedPAC [2], FedRep [3] and our FedCCFA), training the classifier first might cause it to fit the initial features, resulting in gradient attenuation during backpropagation. This will restrict the subsequent extractor's training process and prevent it from optimally learning features. In contrast, when training the entire model together, all layers can simultaneously adapt to the training data, allowing them to synergize effectively. However, in the context of concept drift, decoupled training is crucial because the classifier can effectively adapt to new conditional distribution $P(\mathcal{Y}|\mathcal{X})$ and small gradients will be back-propagated to the extractor. This can stabilize the training of the feature extractor. Besides, the results in Table 1 show that our method still outperforms the majority of baseline methods.
>
> > Q1: What's the need to partition the models into feature extractor and classifier?
>
> A4: In this paper, we focus on real drift, where the conditional distribution $P(\mathcal{Y}|\mathcal{X})$ changes. Since the marginal distribution $P(\mathcal{X})$ is invariant, the representation learning should not be affected by this drift. That is, the feature extractor can be shared across clients to enhance generalization performance. The key to adapt to this drift is the classifier, as it map the representation to the label space (i.e., learning the distribution $P(\mathcal{Y}|\mathcal{X})$). Therefore, decoupling the model into a feature extractor and a classifier can effectively adapt to concept drift while enhancing the generalization performance. The superiority of shared feature extractor can be demonstrated by the better performance than other clustered FL methods under various drift settings.
>
> > Q2: Which is the assumption guaranteeing the clustering of classifier parameters in the parameter space?
>
> A5: Since the classifier is the last fully-connected layer (i.e., a simple linear classifier), the clustering of its parameters makes sense. Please refer to A2 for more details.
>
> > Q3: What's the reason of the reduction in accuracy in no-change condition (i.e., Table 1) and how to mitigate this point?
>
> A6: Please refer to A3 for the reason of this reduction. To mitigate this point, it can be useful to fine-tune the entire model after training the extractor.
>
>
> **References:**
>
> [1] Oh, J., Kim, S., & Yun, S. Y. (2022). FedBABU: Towards enhanced representation for federated image classification. International Conference on Learning Representations.
>
> [2] Xu, J., Tong, X., & Huang, S. L. (2023). Personalized federated learning with feature alignment and classifier collaboration. International Conference on Learning Representations.
>
> [3] Collins, L., Hassani, H., Mokhtari, A., & Shakkottai, S. (2021). Exploiting shared representations for personalized federated learning. In International conference on machine learning (pp. 2089-2099). PMLR.

---

### Author Rebuttal · Authors · 2024-08-05

Dear Reviewers,

We would like to thank all reviewers for providing constructive comments that helped us to improve our paper. We are encouraged that reviews think our paper:

- "The idea of letting the global/local models evolve over time is interesting and relevant" (Reviewer U5Wz),
- "A relatively new and important problem of federated learning under concept drift is addressed" and "Contributions of this work are in the framework perspective and the module/strategy design perspective." (Reviewer HDwb),
- "The paper tackles an important and relatively unexplored problem in federated learning" (Reviewer erp1) and "This work addresses a critical gap in federated learning" (Reviewer uegQ),
- "The experiments are extensive" (Reviewer EQ7w) and "the results show significant performance improvements over existing methods in various concept drift scenarios" (Reviewer erp1)

We have been working diligently on improving the paper on several fronts, addressing your critique. Below, we provide a PDF containing more helpful **pipeline of our FedCCFA** (as suggested by Reviewer HDwb) and **visualization of distance matrix after concept drift** (as suggested by Reviewer erp1). Besides, according to the FAQ of NeurIPS 2024, it is not allowed to add any link in any part of the rebuttal. We will provide our code once this paper is accepted.


Please see our reviewer-specific feedback for more information.

---

### Decision · Program_Chairs · 2024-09-25

**Decision:**

Accept (poster)

**Comment:**

I have read all comments and responses. Reviews appear to be consentaneous, with five scores of 7, 6, 6, 5, and 6. All concerns of reviewers have been fixed. It is recommended to accept this manuscript.